# AutoCode4Math: Learning Autonomous Code Integration for Math LLMs

## Abstract

Recent research on tool integration for math Large Language Models (LLMs) aims to combine complementary strengths of chain-of-thought (CoT) reasoning and code execution. However, we discover a critical limitation: current tool-integrated math LLMs rely on externally dictated instructions to decide whether to use CoT or code, lacking the autonomy to choose the most appropriate method independently. This prompts us to study *Autonomous Code integration* for math LLMs, which enables models to *independently* develop their own methodology-selection strategy in the absence of reliable supervision. To address this challenge, we propose an innovative Expectation-Maximization (EM) formulation that refines the model's decision-making through the exploration of its capabilities. This framework alternates between (a) computing a reference strategy that improves the model's belief over its capabilities through self-exploration, and (b) updating the model based on the refined belief. We further enhance this framework with an efficient implementation, incorporating a novel data synthesis strategy and off-policy reinforcement learning. Extensive experiments demonstrate that our approach, using only a public query set, significantly boosts the performance of existing math LLMs, raising accuracy by nearly 20% to 65.28% on the challenging MATH benchmark, while reducing code executions by up to 65%. Code and data is released in the anonymous repository.

## 1 Introduction

Large Language Models (LLMs) have demonstrated remarkable performance across various domains (Kaddour et al., 2023; Achiam et al., 2023; Dubey et al., 2024; Team et al., 2023; Yang et al., 2024), yet they face significant challenges when addressing complex mathematical problems, such as abstract reasoning and intricate calculations (Gao et al., 2023; Yue et al., 2023; Gou et al., 2023; Li, 2024). To tackle these challenges, two primary paradigms have emerged: *chain-of-thought (CoT) reasoning* (Wei et al., 2022; Yu et al., 2023; Luo et al., 2023) and *the use of external tools*, such as Python programs (Toshniwal et al., 2024; Yue et al., 2023). Both strategies offer distinct yet complementary benefits. CoT is highly effective for semantic analysis, planning, and abstract reasoning, but it often struggles with precise computations. In contrast, leveraging python programs offloads mathematical operations to specialized tools like symbolic solvers or numerical libraries. Nonetheless, this approach has shown vulnerability in effectively translating natural language queries into correct code (Li, 2024).

Recent advancements focus on synergizing these paradigms by developing *tool-integrated reasoning* frameworks, such as TORA (Gou et al., 2023; Shao et al., 2024), HTL (Li, 2024) and DotaMath (Li et al., 2024a), which merge CoT reasoning with code generation to provide more fluid interactions between abstract reasoning and precise computation. However, despite the potential benefits of incorporating external tools, their successful integration presupposes that the model is both adept at using the tool and capable of reasoning effectively about the problem with the code.

This assumption raises concerns about LLMs' capability boundaries of using tools, underscoring the need for more intelligent code integration strategies. Rather than a reactive decision, code integration should be a deliberate decision informed by the model's capabilities, mirroring human-like methodology-selection for problem-solving.

We distill this objective into a central goal: developing *autonomous code integration (AutoCode)* for math LLMs, as illustrated in Fig. 1. Existing tool-integrated math LLMs generally lack the AutoCode capability, because these models are trained to follow an externally imposed methodology that do not adapt to their unique strengths. Therefore, we ask:

*How to enable LLMs to learn their own strategies to select the methodology for math queries, such that it complements the model's inherent capabilities?*

To address this challenge, we introduce a novel Expectation-Maximization (EM) formulation that enables a self-teaching mechanism without relying on external dictations. Specifically, the Expectation step (E-step) computes a reference strategy through self-exploration, representing the belief over its capabilities. The maximization step (M-step) updates the LLM based on the data induced by the new belief. Furthermore, we propose an efficient joint training scheme for this framework, which features a novel data synthesis strategy for offline data curation and efficient off-policy reinforcement learning (RL).

Extensive experiments show that, using *merely public query set*, our method significantly improves existing math LLMs. For instance, our 7B model enhances accuracy on the challenging MATH benchmark (Hendrycks et al., 2021) by nearly 20% and reaches 65.28% accuracy, while reducing up to 65% code executions.

Our contribution is summarized as follows: (1) We identify a critical gap in the existing research on math LLMs, that they struggle with autonomous code integration (AutoCode). (2) We propose a novel EM learning framework that empowers LLMs to develop their own AutoCode strategies, tailored to their inherent capabilities. (3) We showcase a neat and efficient implementation of this EM framework, featuring a novel data synthesis strategy and an efficient off-policy RL training. Extensive results show substantial improvements in both accuracy and computational efficiency.

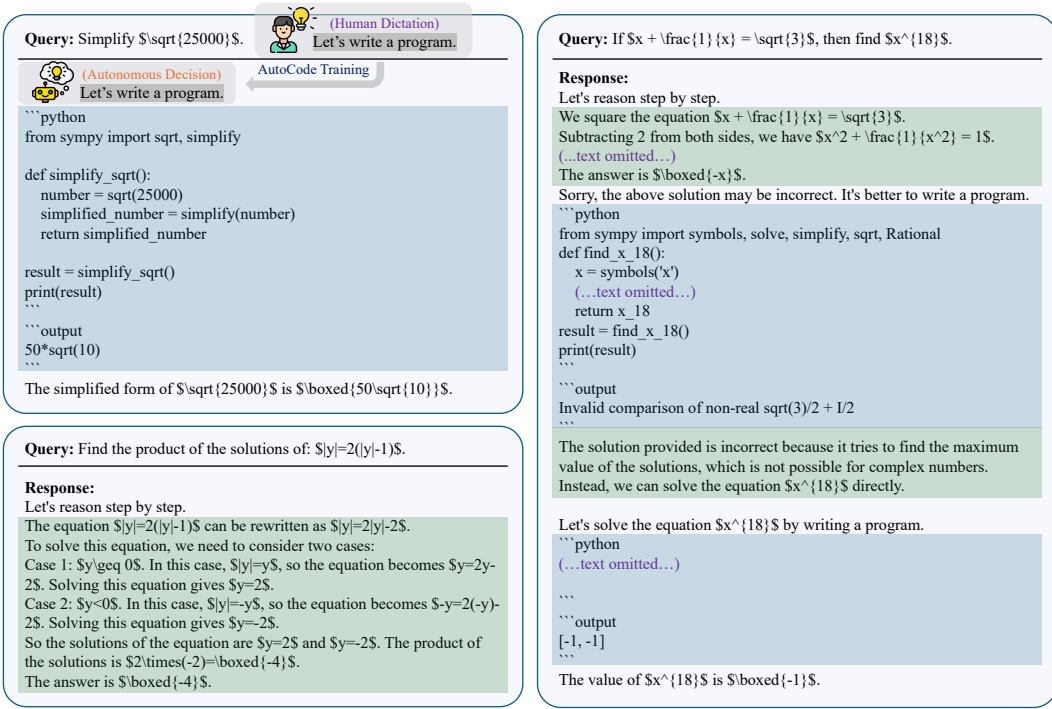

Figure 1: **Examples of Autonomous Code Integration.** Prior models are usually trained on queries with explicit instructions (Top Left). In contrast, our model autonomously learns to choose the appropriate methodology, either CoT or code execution (Left). Additionally, it is capable of performing multiple rounds of autonomous code integration (Right).

## 2 METHODOLOGY

### 2.1 PROBLEM STATEMENT

Recent research have studied tool integration for math LLMs. Specifically, given a math query $x_q \in X_Q$, expressed in natural language, the goal is to generate a solution using a language model (LM), denoted as $\mathcal{M}_\theta$, parameterized by $\theta$. Importantly, the LM is augmented with access to a Python interpreter, allowing it to execute Python code as needed during inference. We are interested in allowing the model to choose the appropriate methodology (denoted as $c$) – either CoT reasoning or code snippets – and generate the corresponding solutions $y_t \in \mathcal{Y}$ based on the preceding context $(x_q, y_{<t})$. When code integration is necessary, the model generates special tokens ```` ```python ```` and ```` ``` ```` to trigger the Python interpreter, after which the execution results are appended to the ongoing context. Both the math query $x_q$ and the response $y_t$ consist of sequences of natural language tokens, with interleaved code integration as necessary (see Fig. 1 for an example).

Our goal is to develop a language model that autonomously integrates code execution for math problem-solving, switching seamlessly between CoT reasoning and code execution when appropriate. Conceptually, such code-integrated reasoning process involves two repeated steps: (a) *Methodology selection.* The model first decides whether to use code execution or CoT reasoning based on the context; (b) *Solution generation.* Based on the chosen methodology, the model generates either executable Python code or CoT rationales.

We note that the methodology selection step can occur either implicitly (via the model's internal states) or explicitly (through verbal cues generated by the model). For example, in our implementation, the model is trained to signal the chosen method at the beginning of its response (see Fig. 1), through verbal cues such as `Let's write a program` for code integration, and `Let's reason step by step` for CoT reasoning. However, it is not required to signal every methodology-selection decisions. For instance, as shown in Fig. 1 (Right), the model can correct code using CoT reasoning without explicit cues.

We formally define our problem as follows. Let $c \in \mathcal{C} = \{0, 1\}$ denote the methodology of CoT and code integration, respectively. Given the math query $x_q$, the model selects a methodology $c$, and generates a corresponding solution $y_a \sim p(y_a|x_q, c; \mathcal{M})$. This decision-making process iterates until the LM predict an end token. Upon termination, the model receives a binary feedback $r = \mathbb{I}(y_a) \in \{0, 1\}$, where the indicator function $\mathbb{I}(y_a)$ evaluates the correctness of the predicted solution $y_a$ by comparing it to the gold answer. For clarity and simplicity, we focus our discussion on a single iteration of the methodology-selection and solution-generation process, which adheres to the following factorization of the language policy: $p_\theta(y_a|x_q) = \sum_c p_\theta(c, y_a|x_q) = \sum_c p_\theta(c|x_q) \cdot p_\theta(y_a|x_q, c)$. And we illustrate in Sec. 2.2 how we handle multiple iterations.

### 2.2 THE EM FORMULATION

**Challenge and Motivation.** The key challenge in achieving autonomous code integration (AutoCode) lies in the absence of reliable supervision for making methodology-selection decisions. Existing math LLMs typically employ supervised fine-tuning (SFT) to empower tool integration (Yue et al., 2023; Wang et al., 2023b; Li et al., 2024a; Gou et al., 2023; Shao et al., 2024; Li, 2024), which relies on externally dictated expert decisions. Training on such data fall short because it fails to dynamically adapt to the model's unique capabilities.

To address this challenge, we draw inspiration from human decision-making. Humans evaluates whether to seek external assistance based on the complexity of the task and their belief in their ability to handle it. This belief is refined through exploration, observation and reflection. For instance, consider solving a complex math problem, such as finding the roots of an equation. A person might attempt manual calculations using roots formula, or resort to a calculator. To form a belief about whether external tools are necessary, humans explore both options and observe the outcomes. Through reflection on such experiences, they recognize their limitations in manual computation results, and develop a belief that relying on a calculator improves accuracy for such problems.

**The EM Framework.** Inspired by this analogy, we propose an Expectation-Maximization (EM)-based approach that emulates the cognitive process of belief refinement and intelligent decision-making, alternating between two key steps (Fig. 2):

Figure 2: **Method Overview.** (Left) shows an overview for the EM framework, which alternates between finding a reference methodology-selection strategy through self-exploration and updating the LLM via self-taught fine-tuning and off-policy reinforcement learning based on synthesized data. (Right) shows an example of the proposed data synthesis strategy.

1. *Self-Exploration (E-step)*: Explore the model's inherent capabilities to identify a reference strategy for methodology selection that maximizes utility.

2. *Self-Refinement (M-step)*: Refine the model's belief about its capabilities under the supervision of the reference strategy, and evolve the model's capabilities through reinforcement.

To address the lack of direct supervision for methodology selection, we treat $c$ as a latent variable. Our goal is to identify the methodology $c$ and corresponding solution $y_a$ that maximizes the likelihood of generating a correct response for a given query. This motivates transforming the maximum value problem into a maximum likelihood problem, which we define as:

$$\mathcal{J}_{\text{MLE}}(\theta) \doteq \log P(r = 1|x_q; \theta) \tag{1}$$

$$= \log \sum_c p_\theta(c|x_q) \sum_{y_a} p_\theta(y_a|x_q, c) \cdot P(r = 1|y_a, c, x_q). \tag{2}$$

The factorization reflects the distinct roles of methodology-selection $p_\theta(c|x_q)$ and the solution-generation $p_\theta(y_a|x_q, c)$ in the LLM's decision-making process.

The EM framework provides an effective solution for maximizing this MLE objective in the presence of latent variables Bishop & Nasrabadi (2006). Intuitively, EM handles the methodology-selection decision $c$ by introducing a reference distribution (strategy) $s(c|x_q)$, analogous to a belief system in human decision-making.

From the MLE objective, we derive the evidence lower bound (derivations detailed in Sec. A.1):

$$\mathcal{J}_{\text{MLE}}(\theta) \geq \mathbb{E}_{s(c|x_q)} \left[ \log \frac{p_\theta(c|x_q) \cdot P(r = 1|c, x_q; \theta)}{s(c|x_q)} \right] \tag{3}$$

$$\doteq \mathcal{J}_{\text{ELBO}}(s, \theta). \tag{4}$$

The EM framework alternates between two-steps.

**E-step.** This step seeks the reference strategy $s(c|x_q)$ by optimizing the following objective:

$$\max_s \mathcal{J}_{\text{ELBO}}(s, \theta) = \min_s D_{\text{KL}} \left( s(c|x_q) \| P(r = 1, c|x_q; \theta) \right). \tag{5}$$

This effectively computes the reference strategy $s(c|x_q)$ as the posterior of methodology $c$ generating correct response for a given query $x_q$, i.e., $P(r = 1, c|x_q; \theta)$. If decision $c$ are more likely to succeed, the reference strategy accordingly assigns higher probability mass to the decision. *By computing the reference strategy, E-step provides effective guidance for methodology-selection in the absence of reliable supervision.*

**M-step.** Using the reference strategy $s(c|x_q)$ from the E-step, the M-step updates the model parameters $\theta$ to maximize the ELBO:

$$\max_\theta \mathcal{J}_{\text{ELBO}}(s, \theta) = \max_\theta \mathbb{E}_{s(c|x_q)} \left[ \mathbb{E}_{p_\theta(y_a|x_q, c)}[R(x_q, y_a)] + \log p_\theta(c|x_q) \right]. \tag{6}$$

The objective comprises two terms focusing on different sub-policies. The first term optimizes the solution-generation policy $p_\theta(y_a|x_q, c)$ for reward maximization, akin to reinforcement learning through exploration. The second term computes a log-loss, essentially supervising the current methodology-selection strategy $p_\theta(c|x_q)$ using the reference strategy.

**Discussion.** Our EM framework mirrors the human cognitive process of belief refinement and intelligent decision-making. During the E-step, the framework engages in a form of *self-exploration by probing its internal model capabilities* to identify a reference strategy. This approach stands in contrast to SFT, which relies on external demonstrations.

In the subsequent M-step, the focus shifts to self-refinement: the model adjusts its methodology selection based on the reference strategy while reinforcing solution generation. This iterative process enables the model to dynamically adapt to its own strengths and weaknesses, meanwhile fostering continuous improvement. Unlike standard Reinforcement Learning (RL) methods, which enhance decision-making through localized exploration around the current policy, *our proposed approach uniquely narrows the policy search space by leveraging the guidance from the reference strategy*.

Beyond addressing the challenge of autonomous code integration, our EM framework is supported by a robust theoretical foundation. It exhibits favorable properties, such as monotonic improvement of the objective function. Specifically, the evidence lower bound (ELBO) objective satisfies the inequality: $\mathcal{J}_{\text{ELBO}}(s_1, \theta_0) \leq \mathcal{J}_{\text{ELBO}}(s_1, \theta_1) \leq \mathcal{J}_{\text{ELBO}}(s_2, \theta_1)$, indicating *a guaranteed progression toward better performance as the model iterates through the EM steps*.

## 2.3 PRACTICAL IMPLEMENTATION

While the EM framework provides a promising solution for AutoCode, its practical application faces two primary challenges. First, the E-step (Eq.5) involves computing the posterior, which is inherently complex. Second, the M-step (Eq.6) introduces fragmented training processes: SFT for methodology selection using *offline* data and RL for solution generation via *on-policy* interactions. This separation not only complicates implementation but is also nonsensical, as both policies share the same parameter space in a unified LLM policy and should not be trained separately.

To address these challenges, *we present a practical implementation that unifies the training of methodology-selection and solution-generation policies, enabling efficient joint optimization*. This streamlined approach consists of a data curation stage and an offline RL training stage. Specifically, the data curation stage produces data for methodology selection and solution generation, combining them into complete responses using two distinct synthesis strategies. These *complete responses* enable a modified M-step objective (Eq. 6) that supports efficient offline training without the need for on-policy interactions. We provide an algorithm diagram in Appendix Sec. B.

### 2.3.1 DATA CURATION

We first elucidate the relationship between solving the E-step for the reference strategy and generating data for this strategy. Then we elaborate on the data synthesis strategies used to combine the training data for methodology-selection and solution-generation.

**E-step and Data Generation.** In the E-step (Eq. 5), computing the exact posterior for the reference strategy is typically complex, often requiring the training of a variational distribution $s_\phi(c|x_q)$ parameterized by separate neural networks $\phi$ (Zintgraf et al., 2019; Wang et al., 2020; 2023a). However, in our implementation, this complexity is mitigated. The reference strategy is only needed to supervise methodology selection $p_\theta(c|x_q)$, not during inference. Given the binary action space $c \in \{0, 1\}$, the posterior $P(r = 1, c|x_q; \theta)$ can be estimated via Monte Carlo simulations, similar to how humans refine beliefs through exploration and observation: to decide whether external assistance is necessary, humans explore the options and adjust their belief based on the outcomes.

Specifically, the posterior is factorized as $P(r = 1, c|x_q; \theta) = p_\theta(c|x_q; \theta) \cdot P(r = 1|c, x_q)$. We estimate the prior through Monte Carlo rollouts:

$$p_\theta(c|x_q; \theta) \approx \frac{1}{K} \sum_{i=1}^{K} \mathbb{I}\{y_i \text{ uses methodology } c\} \tag{7}$$

where $y_i \sim \pi_\theta$ is one of $K$ random rollouts from the LLM generation policy $\pi_\theta$.

The likelihood $P(r = 1|c, x_q; \theta)$ is defined via the Q-function, $Q(x_q, c)$, which evaluates the expected return of methodology choice $c$ given the context of $x_q$:

$$Q(x_q, c) = \mathbb{E}_{p_\theta(y_a|x_q, c)}[R(x_q, y_a)] \approx \frac{1}{K} \sum_{i=1}^{K} r_i. \tag{8}$$

where $R(x_q, y_a)$ represents the reward function. The Q-values can be estimated using $K$ Monte Carlo rollouts per query-methodology pair. We note that the Q-function aligns directly with the likelihood function $P(r = 1|c, x_q; \theta)$, as both both quantify the effectiveness of the query-methodology pair $(x_q, c)$. This enables us to define: $P(r = 1|c, x_q; \theta) \propto Q(x_q, c)$.

Combining these estimates, the reference strategy satisfies $s^*(c|x_q) \propto p_\theta(c|x_q)Q(x_q, c; \theta)$, forming an energy-based distribution,

$$s^*(c|x_q) = \frac{\exp\left(\alpha \cdot p_\theta(c|x_q)Q(x_q, c; \theta)\right)}{Z(x_q)}.$$

where the temperature $\alpha > 0$ controls the sharpness of the distribution, and $Z(x_q)$ is the partition function that normalizes the distribution. *Indeed, the reference strategy represents a softmaxed distribution over methodology choices, from which we sample to generate data for supervised methodology-selection training.*

**Dataset Synthesis.** Our goal for data curation is to collect a dataset where each dataset sample $(x_q, y, r)$ includes a query $x_q$, a response $y$, and a reward $r$. The complete response $y$ is obtained by concatenating sequences of methodology-selection and solution-generation (see Fig. 2 for an example): $y = \text{Concat}(\{(c, y_a)_t\}_{t=1}^T)$. The methodology choices $c$ are sampled from $s^*(c|x_q)$, and the solutions are generated using two inference modes:

- *Dictated:* Explicitly specifying the methodology $c$ in the instruction.
- *Native:* Natively generating complete responses without guiding methodology.

The data for learning methodology-selection and solution-generation is merged into complete responses using two synthesis strategies:

- *Self-Exploratory Synthesis.* We select dictated and native solutions that aligns with the (sampled) reference decision $c^*$, forming single-round or self-correction responses.
- *Self-Reflective Synthesis.* We create multi-round responses by transitioning between methodologies. For example, initial attempts using one methodology (e.g., CoT) may fail, and we hint by a reflective hint to transition to the another methodology (e.g., code) for a successful solution. Formally, the responses are synthesized as $y = \text{Concat}(\{c^-, y_a^-, c^*, y_a\})$. Here $c^-$ denotes an alternative methodology to the reference decision $c^*$, and $y_a^-$ and $c = c^*$ are the corresponding solutions. Detailed examples are illustrated in Fig. 2.

### 2.3.2 EFFICIENT JOINT TRAINING

Using the curated dataset of complete responses, we unify the learning of methodology-selection and solution-generation into a single offline training process. To achieve this, we modify the M-step objective (Eq. 6) by introducing the off-policy policy gradients:

$$\max_\theta \mathbb{E}_{s(c|x_q), \xi(y_a|x_q, c)} \left[O\left(\xi, p_\theta\right) \cdot R\left(x_q, y_a\right) \cdot \log p_\theta\left(y_a|x_q, c\right) + \log p_\theta\left(c|x_q\right)\right], \tag{9}$$

where $\xi(y_a|x_q, c)$ is a behavior policy, $p_\theta(y_a|x_q, c)$ is the improvement policy, and $O(\xi, p_\theta)$ is the off-policy correction term compensating for the discrepancy between these policies.

To stabilize training, we use query-wise reward whitening and employ a clipping scheme on the off-policy correction term following PPO (Schulman et al., 2017): $O(\xi, p_\theta) = \text{Clip}\left(\frac{p_\theta(y_a, c|x_q)}{\xi(y_a, c|x_q)}, 0.8, 1.2\right)$.

This unified process efficiently integrates the training of two sub-policies into a cohesive framework, allowing for a neat implementation of iterative data curation and offline RL.

## 3 EXPERIMENTS

This section addresses the following research questions:

*Q1: Effectiveness in Improving Math Capabilities.* We evaluate how well the proposed method enhances performance based on existing math LLMs, on both in-domain and out-of-domain mathematical benchmarks.

*Q2: Ablations on the proposed components.* We assess the effect of the EM formulation and the self-reflective data synthesis strategy.

*Q3: Analysis of the learned strategies.* We analyze how the learned model improves accuracy, in particular regarding the learned methodology-selection strategies.

**Datasets.** Training with the proposed method requires only a query set, as it leverages self-taught fine-tuning and reinforcement learning that collect data through self-exploration. Nonetheless, it presupposes the model be able to solve math queries using code. Thus, we compile public SFT data that includes CoT reasoning and code solutions for GSM8K (Cobbe et al., 2021) and MATH (Hendrycks et al., 2021) training queries, from MetaMath (Yu et al., 2023), MathInstruct (Yue et al., 2023), OpenMath (Toshniwal et al., 2024) and MMOS (Chen et al., 2024b). Detailed dataset statistics are provided in the appendix.

**Benchmarks and Evaluation Protocols.** We evaluate our approach on two in-domain datasets: GSM8k (Cobbe et al., 2021) and MATH (Hendrycks et al., 2021). To test generalization, we use four out-of-domain benchmarks:AsDIV (Miao et al., 2021), GaokaoMath2023 (Liao et al., 2024), CollegeMath (Tang et al., 2024), OlympiadBench (He et al., 2024). This selection provides a comprehensive test of math reasoning capabilities, ranging from elementary-level to Olympiad-level math problems. We use accuracy with greedy sampling as our primary metric. We evaluate the correctness of predicted responses using released codes from DeepseekMath (Shao et al., 2024) and Qwen2Math (Yang et al., 2024).

**Baselines.** We compare our method against proprietary models like GPT-4 (Achiam et al., 2023) and Claude (Anthropic, 2024), as well as open-source and tool-integrated math LLMs such as Numina-Math (Li et al., 2024b), Mathstral (AI, 2024), Mammoth (Yue et al., 2023), MathCoder (Wang et al., 2023b), ToRA (Gou et al., 2023), DotaMath (Li et al., 2024a), AlphaMath (Chen et al., 2024a), MathGenie (Lu et al., 2024). To examine the effectiveness of the proposed approach on top of existing math LLMs, we consider AutoCode Training on Llama-3.1 (Dubey et al., 2024), Qwen2Math (Yang et al., 2024), and DeepseekMath (Shao et al., 2024). Since the proposed method preconditions on the coding ability for math, we use the aforementioned SFT data to fine-tune Llama-3.1 and Qwen2Math, and term the resulting model as Code4Math-[ModelFamily]. DeepseekMath natively supports writing code to solve math queries.

Due to computational limits, we focus on experiments with 7B models. We report cited performance metrics for most models, and include larger model performances for the readers' reference.

**Implementation Details.** We use $K = 5$ rollouts for response sampling during self-exploration, using nucleus sampling with a temperature of $1.0$ and top-p set to $0.95$. For computing the reference strategy, we set the temperature parameter $\alpha = \infty$, reducing it to a hard-max (greedy selection) of the top-ranking methodology. Regarding learning rate, we set instruction fine-tuning as $2e^{-5}$, the first RL iteration as $1e^{-5}$, and the consecutive iterations as $5e^{-6}$. We use 8 A100(80G) GPUs. For reproduction, we upload code and relevant materials to the anonymous repository Repo.

### 3.1 MAIN RESULTS

Fig. 1 shows the effectiveness of the proposed method. On three family of models, Llama-3.1, Qwen2Math and DeepseekMath, our method our method significantly improves accuracy while

Table 1: **Effectiveness of AutoCode4Math.** The column "Code?" indicates whether code integration is involved, with ★ representing autonomous determination of code integration by the model. †Alphamath employs Beam Search during inference, improving accuracy at the cost of additional test-time computation. We report Pass@1 Accuracy with greedy decoding. For AutoCode models, code rates are provided in parentheses alongside accuracy. The improvement over code-driven inference is highlighted in the colored row, denoted as $\Delta$. The results show that AutoCode4Math significantly improves accuracy by 10.2% while reducing unnecessary code executions up to 66.78% .

| Model | Code? | In-domain | | Out-of-domain | | | |
|---|---|---|---|---|---|---|---|
| | | GSM8K | MATH | GaoKao | Olympiad | College | AsDIV |
| Proprietary Model | | | | | | | |
| GPT-4 (2024-08) (Achiam et al., 2023) | ✗ | 92.9 | 81.1 | 67.5 | 43.3 | 48.5 | 91.3 |
| GPT-4 Code Interpreter | ✓ | 94.2 | 69.7 | 43.6 | | | 92.6 |
| Claude-3 Opus (Anthropic, 2024) | ✗ | 95 | 60.1 | | | | |
| Open-Source Models | | | | | | | |
| Mammoth-70B (Yue et al., 2023) | ✓ | 76.9 | 41.8 | 25.2 | | | 70.2 |
| ToRA-70B (Gou et al., 2023) | ✓ | 84.3 | 49.7 | 31.7 | | | 86.8 |
| MathGenie-70B (Lu et al., 2024) | ✓ | 88.4 | 51.2 | | | | |
| Mathstral-7B (AI, 2024) | ✗ | 84.9 | 56.6 | 46 | 21.5 | 33.7 | |
| Mammoth-Coder-7B (Yue et al., 2023) | ✓ | 59.4 | 33.4 | 15.3 | | | |
| MathCoder-7B (Wang et al., 2023b) | ✓ | 67.8 | 30.6 | | | | |
| ToRA-Code-7B (Gou et al., 2023) | ✓ | 72.6 | 44.6 | 23.9 | | | |
| DotaMath-Deepseek-7B (Li et al., 2024a) | ✓ | 86.7 | 64.8 | | | | 88.5 |
| AlphaMath-Deepseek-7B (Chen et al., 2024a) | ✓+ BS† | 84.1 | 66.3 | 51.4 | | | |
| AutoCode Training | | | | | | | |
| Llama-3.1-Base-8B (Dubey et al., 2024) | ✗ | 53.9 | 17.42 | 13.25 | 3.11 | 6.1 | 74.99 |
| Code4Math-Llama-3.1-8B | ✗ | 77.03 | 43.9 | 25.97 | 6.96 | 15.72 | 79.41 |
| | ✓ | 77.03 (100%) | 43.9 (100%) | 23.38 | 6.81 | 16.93 | 78.96 |
| **AutoCode4Math-Llama3.1** | ★ | 79.83 (40.94%) | 48.7 (59.5%) | 31.69 | 13.19 | 19.91 | 83.02 |
| $\Delta$ | | 2.8↑ (59.06%↓) | 4.8↑ (40.5%↓) | 8.31↑ | 6.38↑ | 2.98↑ | 4.06↑ |
| Qwen2Math-Base-7B (Yang et al., 2024) | ✗ | 80.74 | 51.82 | 43.37 | 21.62 | 24.98 | 90.42 |
| Code4Math-Qwen2 | ✗ | 81.58 | 52.68 | 43.12 | 18.81 | 22.39 | 89.67 |
| | ✓ | 84.68 (100%) | 52.44 (100%) | 44.94 | 20.59 | 26.26 | 90.07 |
| **AutoCode4Math-Qwen2** | ★ | 87.41 (24.4%) | 64.64 (48.9%) | **50.39** | **27.11** | 28.71 | **91.96** |
| $\Delta$ | | 2.73↑ (75.6%↓) | 10.2↑ (51.1%↓) | 5.45↑ | 6.52↑ | 2.45↑ | 1.89↑ |
| DeepseekMath-Instruct-7B (Shao et al., 2024) | ✗ | 81.27 | 45.32 | 42.86 | 14.81 | 34.46 | 90.43 |
| | ✓ | 84.46 (100%) | 56.74 (100%) | 44.68 | 20.44 | 35.49 | 88.22 |
| **AutoCode4Math-DeepSeek** | ★ | **89.38 (6.74%)** | **65.28 (33.22%)** | 47.01 | 23.85 | **36.3** | **91.65** |
| $\Delta$ | | 4.9↑ (93.26%↓) | 8.54↑ (66.78%↓) | 2.33↑ | 3.41↑ | 0.81↑ | 3.43↑ |

reducing code execution frequency. Notably, AutoCode4Math enhances DeepseekMath's performance by 7% on GSM8k from 82.4% to 89.38%, and 20% on Math from 45.04% to 65.28%. Additionally, it reduces code executions by up to 90% on GSM8k and 65% on MATH. Importantly, our method outperforms models like DotaMath without relying on external annotations, and AlphaMath that requires substantial inference-time beam search. However, the proposed method is orthogonal to these approaches, we anticipate further improvements in combination with them.

## 3.2 ABLATION STUDY

The proposed approach relies on two key elements: (1) the EM framework, which factorizes response generation into methodology-selection $p_\theta(c|x_q)$ and solution-generation $p_\theta(y_a|x_q, c)$, and (2) a self-reflective data synthesis strategy that generates multi-round responses.

**Ablation of the EM Formulation.** We compare our method with a baseline using standard RL without explicit methodology-selection, employing an off-policy RL approach for computational parity. The performance curve on the in-domain test sets is shown in Fig. 3. shows that standard RL rapidly converges to suboptimal performance, whereas AutoCode achieves better final accuracy through sustained improvement. This indicates a clear advantage for our EM framework over standard RL.

We also analyze code execution rates for both approaches on in-domain test sets. As Tab. 2 shows, the standard RL approach often introduces more code executions while remaining inferior accuracy. We hypothesize that standard RL focuses solely on outcome optimization, overlooking the role of methodology-selection in improving final accuracy. In contrast, our EM framework enforce its methodology-selection by leveraging the guidance from the reference strategy, which is computed through exploration of its capability boundaries.

Logically, selecting the methodology yielding higher expected utility must in turn improve the final accuracy. We conjecture that the proposed EM framework outperforms standard RL because: the

Table 2: **Ablation Results.** We experiment on top of DeepseekMath (Shao et al., 2024) and Qwen2Math (Yang et al., 2024). We eliminates Self-Reflective Synthesis and EM formulation (reduced to standard RL) from AutoCode4Math. The results reveal the effects of these components in both learning effective methodology-selection and improving accuracy.

| Models | Inference | GSM8k | MATH |
|---|---|---|---|
| AutoCode4Math-Deepseek | Auto | 89.39 (6.75%) | 65.28 (32.22%) |
| wo/ Self-Reflective Synthesis (Iter1) | Auto | 83.09 (2.73%) | 57.92 (23.68%) |
| wo/ EM (RL Iter2) | Auto | 82.94 (0.3%) | 56.74 (29.06%) |
| DeepseekMath-Instruct | Code (100%) | 84.46 | 56.74 |
| AutoCode4Math-Qwen2 | Auto | 87.41 (24.41%) | 64.64 (48.9%) |
| wo/ Self-Reflective Synthesis (Iter1) | Auto | 84.61 (21.23%) | 60.36 (34.48%) |
| wo/ EM (RL Iter2) | Auto | 86.28 (69.6%) | 61.28 (64.66%) |
| Code4Math-Qwen2 | Code (100%) | 84.68 | 52.44 |

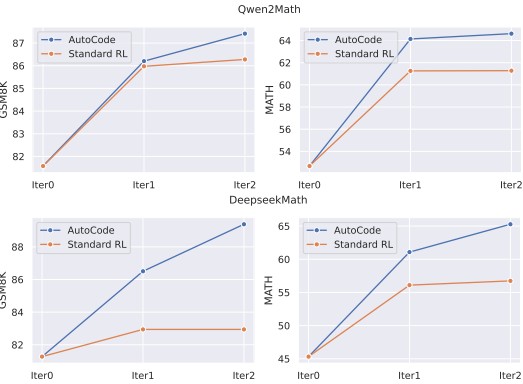

Figure 3: **Performance Convergence.** Experiments are conducted based on Qwen2Math (Top) and Deepseek-Math (Bottom). AutoCode achieves higher accuracy with sustained improvement, while standard RL converge to sub-optimal solutions.

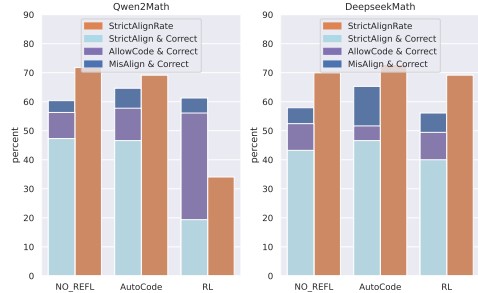

Figure 4: **Analysis of the Learned Strategies.** Correct Responses are classified based on their alignment to the oracle selection, namely, *StrictAlign*, *AllowCode* and *MisAlign*. We show how different categories of alignment contribute to the accuracy in the stacked bars, and include the overall StrictAlign rate in the separate orange bar.

reference methodology-selection strategy helps narrow the policy search space during training. By restricting training samples to those methodology choices with higher utility, the language model can converge more efficiently, benefiting from effective guidance of methodology-selection.

**Ablation of the Self-Reflective Synthesis.** To assess the impact of Self-Reflective Synthesis, we compare with a variant that omits this strategy in the initial RL training (denoted **NO_REFL**). Fig. 2 shows that NO_REFL lags significantly behind AutoCode, demonstrating the importance of multi-round response generation in achieving higher accuracy. This observation points out the inherent connection between performance and multi-round responses. We include a analysis of multi-round responses and showcase success and failure cases in the appendix.

### 3.3 ANALYSIS OF THE LEARNED MODEL.

To understand the effectiveness of the learned methodology-selection strategy, particularly its ability to decide when to use or avoid code, we evaluate its contributions to accuracy. We focus our analysis on the first round of methodology-selection, arguing that this initial decision captures the essential characteristics of the strategy, while subsequent rounds play a minor role in overall accuracy.

We introduce the alignment rate as a key metric to measure how closely the learned strategy aligns with an oracle strategy. The oracle strategy, determined over the test set, dictates that code is used only if it is the sole path to a correct response; otherwise, CoT reasoning is preferred. We classify the correct responses into three categories: (1) *StrictAlign*, where decisions match the oracle exactly, (2)

*AllowCode*, where we consider using code acceptable if both code and CoT yield correct answers, and (3) *MisAlign*, where the model's choice diverges from the oracle.

Fig. 4 shows these alignment categories in stacked bars. The RL model has a significantly lower rate of strict alignment compared to AutoCode (lightblue bar), which is not surprising as the RL model is not explicitly trained for this alignment. In contrast, the proposed EM framework is forced to imitate the behavior of the reference strategy, which computes the oracle selection over training queries. Additionally, regarding code executions, the RL model also tends to favor code more than AutoCode, particularly in the case of Qwen2Math. This is because Qwen2Math has better overall accuracy using code than CoT reasoning, and RL captures this statistical relation. In contrast, the proposed method can implicitly reduces code execution as a side effect, because the oracle selection to imitate during training by design involves minimal code executions.

Interestingly, compared with other models, a significant proportion of correct responses from AutoCode are in the *MisAlign* category. This seems counter-intuitive at first sight, but we note that the oracle choice is determined based on their expected utilities, which means the MisAlign choice can result in correct response at a probability, which is inferior to the oracle choice in the expected sense. This implies that a MisAlign choice can be optimized for final accuracy by improving the corresponding solution-generation capability. Further comparison with NO_REFL suggests that multi-round responses also play a crucial role in this advantage.

# 4   RELATED WORK AND DISCUSSION

**Tool-Integrated Math LLMs.** Math language models (LLMs) have been developed primarily along two major paradigms: Chain-of-Thought (CoT) reasoning and the use of external tools, such as Python programs (Yu et al., 2023; Yue et al., 2023; Toshniwal et al., 2024). Each paradigm offers unique benefits, and recent advancements increasingly seek to combine these approaches for synergistic effects, as seen in methods like Hybrid Instruction Tuning (Yue et al., 2023) and various forms of tool-integrated reasoning (Gou et al., 2023; Li et al., 2024a; Li, 2024; Wang et al., 2023b; Shao et al., 2024). Despite these advancements, existing models still face challenges with autonomous code integration, mainly because they rely on supervised fine-tuning where code integration is suboptimally dictated by external annotators. To address this gap, our work introduces a novel EM framework that avoids using external supervision to facilitate AutoCode capabilities.

**EM for RL.** Expectation-Maximization (EM) has proven effective for maximum likelihood problems involving hidden variables, such as Expert Iteration (Anthony et al., 2017), Iterative Maximum Likelihood (Wu, 2016; Agarwal et al., 2019), Meta-Reinforcement Learning (Zintgraf et al., 2019; Wang et al., 2020), and Adversarial Games (Wang et al., 2023a). In the context of math LLMs, the most relevant works are (Singh et al., 2023) and (Ni et al., 2022), which apply EM-style iterative self-training to math problem-solving. Unlike these approaches, we employ the EM framework specifically to tackle the lack of reliable supervision for autonomous code integration in math LLMs.

**Discussion.** This paper investigates autonomous code integration for math LLMs. To address the challenge of unreliable external supervision, we propose to factorize out the hidden methodology-selection from response generation, and develop a novel EM formulation. The EM framework alternates between computing a reference strategy for methodology-selection through self-exploration and updating language model based on the reference guidance. This approach supports an efficient joint training scheme that allows for holistic offline data collection coupled with RL training. Our extensive experiments demonstrate the effectiveness of the proposed method, and our ablation studies further elucidate the properties of the learned model.

However, there are several limitations and areas for future work regarding AutoCode4Math. First, the generalization of methodology-selection depends significantly on the quality of the collected query set. Further research is needed to understand what characteristics of queries contribute to effective generalization. Second, we did not extensively explore the influence of hyperparameters related to RL iterations, such as dataset size and the number of iterations, in the current version. We are actively working on this. Third, as this is a preliminary work in autonomous code integration, we have not yet investigated alternative approaches for decision routing, such as using Mixture-of-Experts (MoEs) (Masoudnia & Ebrahimpour, 2014), and we not yet fully understand the fundamental reason why EM outperforms RL. These areas present important directions for advancing AutoCode capabilities in math LLMs.

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

# A APPENDIX

## A.1 DERIVATION OF THE EM

We first model RL as a maximum likelihood problem. Specifically, we consider 'maximizing expected return' as 'maximizing the likelihood of observing a correct response', this is equivalent to maximizing the following log-likelihood,

$$\max_{\theta} \log P(r = 1|x_q; \theta) \tag{10}$$

$$= \max_{\theta} \log \sum_c p_{\theta}(c|x_q) \sum_{y_a} p_{\theta}(y_a|x_q, c) \cdot P(r = 1|y_a, c, x_q) \tag{11}$$

$$\doteq \mathcal{J}(\theta), \tag{12}$$

where we factorize the language model as $p_{\theta}(y_a|x_q) = \sum_c p_{\theta}(c|x_q) p_{\theta}(y_a|x_q, c)$.

Since the variable $c$ is unobservable for lack of reliable supervision, we resort to the EM framework. Specifically, we treat $c$ as a hidden variable, and introduce a proposal distribution $s(c|x_q)$, which represents the belief of $c$ given the fixed LLM $\mathcal{M}_{\theta}$ and the data evidence. We have the following derivations,

$$\mathcal{J}(\theta) = \log \sum_c s(c|x_q) \cdot \frac{p_{\theta}(\cdot|x_q)}{s(\cdot|x_q)} \cdot \sum_{y_a} p_{\theta}(y_a|x_q, c) \cdot P(r = 1|y_a, c, x_q) \tag{13}$$

$$\geq \sum_c s(c|x_q) \left[ \log \frac{p_{\theta}(c|x_q) \cdot \sum_{y_a} p_{\theta}(y_a|x_q, c) \cdot P(r = 1|y_a, c, x_q)}{s(c|x_q)} \right] \tag{14}$$

$$= \mathbb{E}_{s(c|x_q)} \left[ \log \frac{p_{\theta}(c|x_q) \cdot P(r = 1|c, x_q; \theta)}{s(c|x_q)} \right] \tag{15}$$

$$\doteq \mathcal{J}_{\text{ELBO}}(s, \theta), \tag{16}$$

where $P(r = 1|c, x_q; \theta)$ denotes the likelihood of generating correct responses given $(x_q, c)$ following the solution-generation policy $p_{\theta}(y_a|x_q, c)$.

In the E-step, we are essentially minimizing a KL divergence,

$$\max_s \mathcal{J}_{\text{ELBO}}(s, \theta) = \min_s \text{D}_{\text{KL}} \left( s(c|x_q) \,|s^*(c|x_q) \right), \tag{17}$$

where the minimizer is $s^*(c|x_q) \propto p_{\theta}(c|x_q) \cdot P(r = 1|c, x_q; \theta)$. Intuitively, the belief over the methodology $c$ following a posterior update: it is based on the prior of the current model $p_{\theta}(c|x_q)$ and the likelihood of data $P(r = 1|c, x_q; \theta)$. The optimal methodology-selection strategy assigns higher probability to method $c$ if following the current LLM it has higher likelihood to generate correct responses or higher prior over it.

To compute the optimal strategy, we define $P(r = 1|c, x_q; \theta) = \frac{\exp(\alpha \cdot p_{\theta}(c|x_q) Q^{\theta}(x_q, c))}{Z(x_q)}$ as an energy-based distribution, where the negative energy function evaluates the expected return induced by the

current solution-generation policy: $Q(x_q, c; \theta) = \mathbb{E}_{p_\theta(y_a|x_q,c)}[R(x_q, y_a)]$, $\alpha > 0$ is a temperature controlling the sharpness of the distribution. Then the minimizer can be computed by enumerating over $c$.

In the M-step, we optimize the following,

$$\max_\theta \mathcal{J}_{\text{ELBO}}(s, \theta) = \max_\theta \mathbb{E}_{s(c|x_q)}[\log P(r = 1|c, x_q; \theta)] - \text{D}_{\text{KL}}\left(s(c|x_q) \,|p_\theta(c|x_q)\right) \quad (18)$$

$$= \max_\theta \mathbb{E}_{s(c|x_q)}[Q(x_q, c; \theta)] - \text{D}_{\text{KL}}\left(s(c|x_q) \,|p_\theta(c|x_q)\right), \quad (19)$$

which maximizes the expected return while imitating the optimal strategy.

## B  THE EM ALGORITHM DIAGRAM

---
**Algorithm 1** AutoCode4Math EM Learning

---
1: **Input:** Query set $\mathcal{Q}$
2: **Output:** Parameters $\theta$
3: Initialize parameters from base model $\theta^{(0)}$
4: $t \leftarrow 0$
5: **repeat**
6:     // **E-step: Find the reference strategy** $s^*(c|x_q)$
7:     Perform $K$ random rollouts for each query to collect the query-methodology-responses dataset $\text{D}_{\text{Dictated}}$ and $\text{D}_{\text{Native}}$. Store the outcome $r$, and the log-likelihoods.
8:     Using $\text{D}_{\text{Dictated}}$, compute the Q-values for $x_q \in \mathcal{Q}, c \in \{0, 1\}$ according to equation 8.
9:     Take hard-max or sample from the soft-max of the Q-values to obtain $c^* \sim s(c|x_q)$.
10:     Synthesize complete trajectories to obtain D, using self-exploratory and self-reflective synthesis, based on the reference selection $c^*$ for each query, $\text{D}_{\text{Dictated}}$ and $\text{D}_{\text{Native}}$.
11:     // **M-step: Update the LLM** $p_\theta$
12:     Optimize the LLM according to equation 9, using the dataset D.
13:     $t \leftarrow t + 1$
14: **until** convergence
       **return** $\theta^{(t)}$

---

## C  ADDITIONAL EXPERIMENTS

We aim to answer the following research questions:

*Q4: Effectiveness in Autonomous Code Integration. We evaluate how our models and baselines perform when required to perform autonomous code integration. Specifically, we investigate the effectiveness of their methodology-selection and its relation to the final performance.*

**Baselines.** We compare our approach with several models that natively support both code and Chain-of-Thought (CoT) responses for math queries: GPT-4, Mammoth-70B trained using Hybrid Instruction Tuning, and DeepseekMath-Instruct-7B trained using tool-integrated reasoning annotations.

**Evaluation Metrics.**

- **Final Pass@1 Accuracy of the Complete Response**
  - "Autocode": The LLM autonomously decides the methodology. For baselines without AutoCode Training, a four-shot prompt template (Appendix D.3) is used.
  - "Code": The LLM is explicitly prompted to generate a code response. For GPT-4, a four-shot template is applied. For other baselines, we use their native zero-shot templates.
  - "CoT": The LLM is explicitly prompted to use CoT reasoning with native templates.

- **Autocode CodeRate and Improvement Over Best Dictation**
  - **CodeRate** reflects the reduction in code usage compared to dictated code prompting.

Table 3: **Effectiveness of Autonomous Code Integration.** We compare our models with representative baselines when they are required to make autonomous decisions on methodology. All metrics are represented in percentage. The results show that existing models that support code integration for math reasoning significantly degrades performance when required AutoCode inference (as shown in the Autocode Improvement column). Our model significantly improves over its dictated inference by improving methodology-selection.

| | Acc | CodeRate | Improvement | Sel. mAcc | Acc /w Cor. Sel. | CoT Acc | Code Acc | CoT Sel. Acc | Code Sel. Acc |
|---|---|---|---|---|---|---|---|---|---|
| GPT-4 | 74.16 | 10.8 | -2.5 | 50.51 | 89.27 | 76.66 | 72.22 | 100 | 1.03 |
| Mammoth-70B | 31.46 | 83.64 | -5.76 | 64.87 | 77.62 | 16.66 | 37.22 | 39.08 | 90.66 |
| DeepseekMath-Instruct-7B | 45.72 | 12.94 | -11.54 | 64.70 | 77.77 | 45.32 | 57.26 | 97.62 | 31.79 |
| Ours-Qwen2-7B | 64.64 | 48.9 | **+4.68** | **88.25** | 96.35 | 51.92 | 59.94 | 89.16 | 87.34 |
| Ours-Deepseek-7B | 65.28 | 33.22 | **+5.04** | **86.53** | 94.21 | 43.82 | 60.24 | 78.17 | 94.90 |

- **Accuracy Improvement over best performance** of either CoT or code dictation reflects the improvement of AutoCode over the native dictated inference.

- **Methodology-Selection Accuracy and Its Connection to Final Accuracy**

  - Ground-Truth Labels: Methodology-selection is treated as a binary classification task. The classification label is derived by performing 10 Monte Carlo rollouts per query with controlled methodologies. The optimal methodology is chosen based on higher expected correctness.

  - Imbalanced Classification: Since model capabilities differ, reference decisions are imbalanced (e.g., GPT-4 strongly prefers CoT, with only 7.45% of queries requiring coding). We report **mean accuracy across CoT-preferred and code-preferred queries** (Selection mAcc), along with per-class accuracy (in the last 3 and 4 columns).

  - Connection to Final Accuracy: Correct methodology-selection does not always guarantee a correct response due to prompt context mismatches and greedy decoding. We report **Pass@1 accuracy within correct selections** to evaluate how proper methodology-selection directly contributes to correct responses.

As shown in Tab. 3:

**Effectiveness in Methodology-Selection.** AutoCode Training significantly improves methodology-selection accuracy, outperforming baseline models by over 20% (compare row 3 with row 5). In particular, GPT-4 exhibits low mean accuracy (50.51%) due to its strong bias for CoT. If GPT-4 intelligently selected methodologies, it could achieve an additional 7.62% gain in accuracy for queries requiring code responses.

**Accuracy Improvements and Code Reduction.** AutoCode Training enables our Deepseek-based model to achieve a 5.04% accuracy improvement compared to the best dictated inference (code prompting) while reducing code usage by up to 66%. Similar trends are observed with the Qwen2Math base model. In contrast, baseline models experience substantial accuracy drops when attempting autonomous methodology-selection. For example, DeepseekMath loses 11.54% accuracy when self-selecting methodologies (row 3).

**Connection Between Methodology-Selection and Final Accuracy.** Baseline models surpass random selection (50%) in methodology-selection accuracy but fail to improve final accuracy over best dictation. This is accredited to **the gap between methodology-selection and final correctness**: *better methodology-selection is not always accompanied with correct final response, due to the greedy decoding and mismatch in prompt context. However,* **our proposed EM-based joint training significantly bridge this gap**: it improves Pass@1 accuracy within correct selections, achieving rates as high as 95% (rows 4 and 5). This success is due to training the LLM with complete responses, optimizing both methodology-selection and final correctness jointly.

# D  STATISTICS AND ANALYSIS

We analyze the AutoCode capabilities of the existing tool-integrated LLMs, GPT-4o and Deepseek-Math in Tab. 4. We prompt AutoCode capbilities using the instruction shown in Sec. E.3. Both models show clear preference toward using CoT reasoning, with less than 10% code executions. It is also counter-intuitive at first sight why using very small portion of code executions can result in significant performance degradation – in the case of GPT-4o, only 1% code executions result in

6% decrease in accuracy. We conjecture that this owes to the sensitivity to the AutoCode few-shot prompting. Rarely seeing such demonstrations make the models prone generate wrong responses, compared to standard clean few-shot prompting.

We list the collected public SFT data in Tab. 5.

We show the distribution of multi-round responses of the model AutoCode4Math-Deepseek in Tab. 6.

Table 4: **Analysis of existing tool-integrated math LLMs.** We find that two state-of-the-art tool-integrated system, GPT-4o and DeepseekMath, struggle with AutoCode. When given few-shot demonstrations for choosing between CoT and code to solve given math queries, they show clear preference of using CoT (less than 10% code executions). This reveals that these model cannot decide on their own what methodologies to choose for a given math query. In addition, we observe that there is significant performance degradations when prompted for AutoCode: only 1% code executions lead to 6% accuracy decrease in the case of GPT-4o. This reflects that their performance are sensitive to the given few-shot prompting.

| Model | CoT (Acc/Code Rate) | Code (Acc/Code Rate) | AutoCode-4shots (Acc/Code Rate) |
|---|---|---|---|
| GPT-4o | 77.04 / 0 | 57.34 / 97.58 | 71.54 / 1.24 |
| Deepseek-Math-Instruct | 46.92 / 0 | 46.26 / 99.92 | 41.46 / 5.18 |
| Mammoth-Mistral-7B | 19.18 / 0 | 40.2 / 99.38 | 37.58 / 95.5 |
| Mammoth-7B | 10.7 / 0 | 29.1 / 99.96 | 23.86 / 95.4 |
| Mammoth-13B | 12.7 / 0 | 31.4 / 98.96 | 22.54 / 60.68 |
| Mammoth-70B | 20.6 / 0.02 | 40.58 / 99.62 | 33.4 / 83.66 |

Table 5: **Public SFT Data Used in this Work.** We collect public query set for AutoCode Training. After Deduplication, the total amount of query used is 119K. If the base model is not trained to write code for math problems, we use the SFT annotations associated with the above queries.

| Dataset | Size | Unique Queries | CoT Solutions | Code Solutions |
|---|---|---|---|---|
| Openmath (Toshniwal et al., 2024) | 129917 | 70002 | 25116 | 104801 |
| Math-Instruct (Yue et al., 2023) | 237781 | 219607 | 188644 | 49137 |
| Metamath Yue et al. (2023) | 285000 | 161337 | 285000 | 0 |
| MMOS (Chen et al., 2024b) | 134610 | 69007 | 0 | 134610 |

Table 6: **Distribution of Multi-Round Responses.**

| Model | Dataset | Queries | Round1 | Round2 | Round3 |
|---|---|---|---|---|---|
| Deepseek-math-AutoCode | MATH | 5000 | 1252 | 3719 | 29 |
| | GSM8k | 1319 | 1013 | 304 | 2 |

# E EXAMPLES

## E.1 SUCCESS CASES

Listing 1: Success case for multi-round.

```
Round 1:
Question: Henry and 3 of his friends order 7 pizzas for lunch.
    Each pizza is cut into 8 slices. If Henry and his friends want
     to share the pizzas equally, how many slices can each of them
     have?
Solution:
There are 7 pizzas in total, and each pizza is cut into 8 slices.
    So, there are 7 * 8 = 56 slices in total.
Henry and his 3 friends want to share the pizzas equally, so they
    will each have 56 / 4 = 14 slices.
```

Each of them can have 14 slices.
The answer is $\\boxed{14}$.

------------------

Round2:
Question: Henry and 3 of his friends order 7 pizzas for lunch.
    Each pizza is cut into 8 slices. If Henry and his friends want
    to share the pizzas equally, how many slices can each of them
    have?
Solution:
```python
def pizza_slices_per_person():
"""Henry and 3 of his friends order 7 pizzas for lunch. Each pizza
    is cut into 8 slices. If Henry and his friends want to share
    the pizzas equally, how many slices can each of them have?"""
total_pizzas = 7
slices_per_pizza = 8
total_slices = total_pizzas * slices_per_pizza
people = 4 # Henry and his 3 friends
slices_per_person = total_slices // people # Use floor division to
    get an integer result
return slices_per_person

slices_per_person = pizza_slices_per_person()
print(slices_per_person)
```
```output
14
```
Each of them can have $\boxed{14}$ slices.

------------------

Round3:
Question: Peter has 4 boxes with the same number of chocolate bars
    in each, while Martha has 7 boxes with the same number of
    chocolate bars in each. If Peter and Martha have totals of 64
    and 56 chocolate bars respectively, how many more chocolate
    bars does Peter have in each box than Martha?
Solution:
If Peter has 4 boxes and Martha has 7 boxes, then Martha has 7/4 =
    1.75 times as many boxes as Peter.
If Martha has 56 chocolate bars and Peter has 64 chocolate bars,
    then Peter has 64/56 = 1.142857 times as many chocolate bars
    as Martha.
If Martha has 1.75 times as many boxes as Peter, and Peter has
    1.142857 times as many chocolate bars as Martha, then Martha
    has 1.75/1.142857 = 1.5 times as many chocolate bars in each
    box as Peter.
Therefore, Peter has 1.5 - 1 = 0.5 more chocolate bars in each box
    than Martha.
The answer is $\boxed{0.5}$.
Sorry, the above solution may be incorrect. It's better to write a
    program.
```python
def chocolate_bars_difference():
"""Peter has 4 boxes with the same number of chocolate bars in
    each, while Martha has 7 boxes with the same number of

```
        chocolate bars in each. If Peter and Martha have totals of 64
        and 56 chocolate bars respectively, how many more chocolate
        bars does Peter have in each box than Martha?"""
peter_total = 64
martha_total = 56
peter_boxes = 4
martha_boxes = 7
peter_bars_per_box = peter_total / peter_boxes
martha_bars_per_box = martha_total / martha_boxes
difference = peter_bars_per_box - martha_bars_per_box
return difference

difference = chocolate_bars_difference()
print(difference)
```
```output
8.0
```
Peter has $\boxed{8}$ more chocolate bars in each box than Martha.

---

### E.2 FAILURE CASES

Listing 2: Failure case for multi-round.

```
Round 1:
Question: The solution to the inequality
```
$$\frac{x + c}{x^2 + ax + b} \le 0$$ is $x \in (-\infty,-1) \cup [1,2).$ Find $a + b + c.$
```
Solution:
Let's reason step by step.
The inequality is equivalent to
```
$$(x + c)(x^2 + ax + b) \le 0.$$ Since the roots of $x^2 + ax + b$ are $-1$ and $1,$ we can write
$$x^2 + ax + b = (x + 1)(x - 1) = x^2 - 1.$$ Then
$$(x + c)(x^2 + ax + b) = (x + c)(x^2 - 1) = (x + c)(x + 1)(x - 1).$$ The inequality is satisfied when $x \in (-\infty,-1) \cup [1,2),$ so
$$(x + c)(x + 1)(x - 1) \le 0$$ when $x \in (-\infty,-1) \cup [1,2).$ This means that one of $x + c,$ $x + 1,$ or $x - 1$ is nonpositive, and the other two are nonnegative.

If $x + c \le 0,$ then $x \le -c.$ Since $x \in [1,2),$ we must have $-c \ge 2,$ or $c \le -2.$

If $x - 1 \le 0,$ then $x \le 1.$ Since $x \in (-\infty,-1),$ we must have $-1 \le 1,$ which is always true.

If $x + 1 \le 0,$ then $x \le -1.$ Since $x \in [1,2),$ we must have $-1 \le 1,$ which is always true.

Therefore, the solution is $x \in (-\infty,-c] \cup [1,2).$ Since $x \in [1,2),$ we must have $-c \ge 2,$ or $c \le -2.$

Hence, $a + b + c = 0 - 1 + (-2) = \boxed{-3}.$
The answer is $\boxed{-3}.$

------------------
```

```
Round2:
Question: The girls are trying to raise money for a carnival. Kim
    raises $320 more than Alexandra, who raises $430, and Maryam
    raises $400 more than Sarah, who raises $300. How much money,
    in dollars, did they all raise in total?
Solution:
```python
def total_money_raised():
    """The girls are trying to raise money for a carnival. Kim
        raises $320 more than Alexandra, who raises $430, and
        Maryam raises $400 more than Sarah, who raises $300. How
        much money, in dollars, did they all raise in total?"""
    alexandra_money = 430
    kim_money = alexandra_money + 320
    sarah_money = 300
    maryam_money = sarah_money + 400
    total_money = alexandra_money + kim_money + sarah_money +
        maryam_money
    return total_money

total_money = total_money_raised()
print(total_money)
```
```output
2180
```
The girls raised a total of $\boxed{2180}$ dollars.

------------------

Round3:
Question: The digits from 1 to 6 are arranged to form a six-digit
    multiple of 5. What is the probability that the number is
    greater than 500,000? Express your answer as a common fraction
    .
Solution:
Let's reason step by step.
To form a six-digit multiple of 5, the last digit must be either 0
    or 5.
Since the number must be greater than 500,000, the last digit must
    be 5.
The remaining five digits can be arranged in $5!$ ways.
Therefore, the probability is $\frac{5!}{6!} = \boxed{\frac{1}{6}}
    $.
The answer is $\boxed{\frac{1}{6}}$.
Based on the above reasoning, we can verify by writing a program.
```python
from sympy import Rational

def probability_greater_than_five_million():
    """The digits from 1 to 6 are arranged to form a six-digit
        multiple of 5. What is the probability that the number is
        greater than 500,000? Express your answer as a common
        fraction."""
    # Total number of arrangements
    total_arrangements = 6*5*4*3*2*1

    # Number of arrangements that start with 5 or 6
```

```
    valid_arrangements = 2*5*4*3*2*1

    # Number of arrangements that start with 1
    invalid_arrangements_1 = 1*5*4*3*2*1

    # Number of arrangements that start with 2
    invalid_arrangements_2 = 1*4*3*2*1

    # Number of arrangements that start with 3
    invalid_arrangements_3 = 1*3*2*1

    # Total number of invalid arrangements
    invalid_arrangements = invalid_arrangements_1 +
        invalid_arrangements_2 + invalid_arrangements_3

    # Calculate the probability
    probability = Rational(valid_arrangements, total_arrangements)

    return probability

result = probability_greater_than_five_million()
print(result)
```
```output
1/3
```
The probability that the number is greater than 500,000 is $\boxed
    {\frac{1}{3}}$.
```

---

### E.3 AUTOCODE 4SHOTS

Listing 3: Autocode 4shots example.

```
User: Simplify $\sqrt{25000}$.",
Assistant:
Let's write a program.
```python
from sympy import sqrt

def simplify_sqrt():
    """Simplify $\sqrt{25000}$."""
    simplified_sqrt = sqrt(25000)

    return simplified_sqrt

result = simplify_sqrt()
print(result)
```
User:The function $f(x)$ satisfies \[f(x + f(x)) = 4f(x)\]for all
    $x,$ and $f(1) = 4.$ Find $f(21).$
Assistant:
Let's reason step by step.
We can start by plugging in $x = 1$ into the given equation. This
    gives us
\[f(1 + f(1)) = 4f(1).\]
Since we know that $f(1) = 4$, we can substitute this into the
    equation to get
```

\[f(1 + 4) = 4 \cdot 4.\]
Simplifying, we have
\[f(5) = 16.\]
Next, we can plug in $x = 5$ into the given equation. This gives
    us
\[f(5 + f(5)) = 4f(5).\]
Since we know that $f(5) = 16$, we can substitute this into the
    equation to get
\[f(5 + 16) = 4 \cdot 16.\]
Simplifying, we have
\[f(21) = 64.\]
Therefore, the value of $f(21)$ is $\\boxed{64}$.
The answer is $\\boxed{64}$.
User:Find the matrix that corresponds to reflecting over the $x$-
    axis.
Assistant:
Let's reason step by step.
The reflection of a point $(x, y)$ over the $x$-axis is $(x, -y)$.
Thus, the matrix that corresponds to reflecting over the $x$-axis
    is $\\boxed{\begin{pmatrix} 1 & 0 \\ 0 & -1 \end{pmatrix}}$.
The answer is $\\boxed{\begin{pmatrix}1&0\\0&-1\end{pmatrix}}$.
User:If $a$ is three times $b$, $b$ is twice $c$, and $c$ is four
    times $d$, what is the value of $\frac{a\cdot c}{b\cdot d}$?
Assistant:
Let's write a program.
```python
from sympy import symbols, simplify

def ratio_of_products():
    a, b, c, d = symbols('a b c d')

    # Given relations
    a = 3 * b
    b = 2 * c
    c = 4 * d

    # Substitute the relations into the expression
    expr = a * c / (b * d)
    expr = expr.subs({a: 3 * b, b: 2 * c, c: 4 * d})

    # Simplify the expression
    simplified_expr = simplify(expr)

    return simplified_expr

result = ratio_of_products()
print(result)
```

