# OpenReview forum: "AutoCode4Math: Learning Autonomous Code Integration for Math LLMs"
_ICLR.cc/2025/Conference — Submitted to ICLR 2025_

### Official Review · Reviewer_nGHB · 2024-10-31

**Soundness:** 2
**Presentation:** 1
**Contribution:** 3
**Rating:** 5
**Confidence:** 3

**Summary:**

The paper, titled "AutoCode4Math: Learning Autonomous Code Integration for Math LLMs," addresses the limited ability of math-focused language models to make independent decisions on whether to use chain-of-thought (CoT) reasoning or code execution for problem-solving.
Generally, such models follow externally given instructions for choosing amongst them, narrowing their ability to operate autonomously while the integration of the best option will be done for answering the query on mathematics. To tackle it, the authors build an Expectation-Maximization framework that offers the ability for the LLM to learn autonomous code integration by furthering decision-making capacity through self-exploration. In this regard, the model alternates between self-teaching phases, E-step, after which an update derived from refined beliefs is set, M-step, to form an optimized methodology selection strategy. Experiments show that such an approach results in a significant accuracy gain when tested with lower code-execution requirements: an improved performance of nearly 20% on the MATH benchmark, with virtually a 65% reduction in code executions.

**Strengths:**

The study makes a significant contribution by addressing how models can autonomously choose between Proof-of-Thought (PoT) and Chain-of-Thought (CoT) approaches during mathematical problem-solving. This research identifies an important gap in current language models' ability to self-direct their reasoning strategies. Through their experiments, the authors demonstrate that incorporating autonomous decision-making in training leads to measurable performance improvements. The results validate that self-directed methodology selection, driven by expectation maximization, enhances mathematical problem-solving capabilities.

**Weaknesses:**

# Motivation for Using Expectation-Maximization (EM)

The paper introduces Expectation-Maximization (EM) to autonomously select between Chain of Thought (CoT) or code methodologies for solving math queries without external supervision. However, the motivation for this choice of EM remains unclear in several respects:
- Lack of Comparative Analysis: While the paper suggests EM is superior to standard reinforcement learning (RL), it does not provide a rigorous comparative analysis or an explanation of its advantages over other optimization techniques. The narrative suggests that EM holds unique benefits, but it is unclear why it is preferred over RL or simpler alternatives like supervised fine-tuning with reinforcement. Additional justification is needed to establish EM’s unique suitability for method selection in this context.
- Efficiency Assumptions: The use of EM is partly based on the assumption that it narrows the policy search space, thereby enhancing efficiency. However, there is no empirical evidence provided to demonstrate that EM’s efficiency leads to actual performance gains in selecting between CoT and code for complex queries. Without comparative data, it is difficult to assess whether this efficiency is observed in practice or remains theoretical in this case.
# Insufficient Comparison with Baselines for Model Selection
The effectiveness of AutoCode4Math’s approach to method selection would benefit from more detailed comparisons with existing baselines:
- Baseline Limitations: While current baseline models (e.g., Qwen2Math, GPT-4o, and DeepseekMath) primarily employ CoT over code for math queries, the lack of detailed performance metrics for these baselines limits the reader’s ability to assess AutoCode4Math’s advantages. Controlled comparisons with these models would help substantiate claims about AutoCode’s effectiveness in methodology selection.
- Effectiveness of Autonomous Decision-Making: The paper highlights AutoCode4Math’s advantage in reducing code executions and improving accuracy, but lacks direct performance comparisons with established models like GPT-4 and Mammoth. Quantitative evidence of accuracy improvements tied to better method selection would strengthen the claims of AutoCode4Math’s benefit.
- Comparisons with Standard RL Techniques: Although the paper briefly mentions an offline RL baseline, it does not detail the data, algorithm specifics, or other critical aspects of that experiment. A comparison with rejection-sampling and preference-learning-based approaches would further contextualize AutoCode4Math’s methodology and showcase its relative effectiveness.

**Questions:**

### Motivation for Using EM
1. **Why is EM chosen over alternative methods like standard reinforcement learning (RL) or supervised fine-tuning?**
   - What specific advantages does EM bring to methodology selection that simpler or more commonly used techniques cannot provide?

2. **How does EM uniquely contribute to the narrowing of the policy search space?**
   - Could you provide empirical evidence or a theoretical basis for how EM improves efficiency in selecting between Chain of Thought (CoT) and code-based approaches?

3. **What is the theoretical or experimental basis for assuming that EM will reduce computational load or enhance efficiency in methodology selection?**
   - How does EM’s efficiency in narrowing the search space compare to RL or other optimization methods in this particular task?

### Comparison with Baselines for Model Selection
4. **How does AutoCode4Math perform compared to current baselines, specifically in terms of method selection effectiveness?**
   - Could you provide a detailed, controlled comparison with baseline models like Qwen2Math, GPT-4o, and DeepseekMath, showing how frequently and accurately AutoCode4Math selects CoT versus code?

5. **What improvements in accuracy or reduction in code execution does AutoCode4Math achieve over established models such as GPT-4 and Mammoth?**
   - Are there quantitative metrics available that directly correlate the model’s method selection accuracy with overall performance improvements?

6. **Could you expand on the offline RL baseline and provide details on the data, algorithm, and setup of this experiment?**
   - What comparisons are made with other RL methods or preference-learning-based approaches? How does AutoCode4Math fare against these methods, particularly in terms of accuracy and computational efficiency?

---

> ### Author Response · Authors · 2024-11-17
> **Rebuttal by Authors (Part1)**
>
> Dear Reviewer #nGHB,
>
> We thank the reviewer for the valuable feedback and constructive suggestions. We have carefully addressed each of your questions and detailed our responses below.
>
> **Motivation for Using EM.**
>
> **1. Response to the weakness comments on Lack of Comparative Analysis and Q1.**
>
>
> In the current paper, the reasons for choosing EM over alternatives are scattered, e.g., line 056, line 122, line 182, line 259. To address this, we have summarized the motivation clearly below and will ensure these reasons are explicitly clarified in the revised manuscript.
>
> => _Q1: Why is EM chosen over alternative methods like standard reinforcement learning (RL) or supervised fine-tuning?_
>
> The motivation for our EM framework lies in the need for math LLMs to autonomously develop effective methodology-selection strategies that complement their inherent strengths (line 059). The central challenge is the lack of reliable supervision for methodology-selection (line 182).
>
> *   Limitations of SFT: Supervised Fine-Tuning (SFT) relies on external expert annotations. These data fail to dynamically adapt to the model’s specific strengths (line 057).
>
> *   Limitations of Standard RL. While standard RL can explore the policy space to find high-reward regions, it is limited to local explorations around the current policy, which is less efficient in policy training. This is supported by our experimental results (Figure 4).
>
> *   Advantages of the proposed EM approach.
>
>     *   **Direct Supervision for Methodology-Selection:** We reformulate the problem as maximum likelihood estimation with latent variables, naturally leading to a theoretically sound EM solution. This approach addresses the challenge of lacking reliable supervision by generating reference decisions during the E-step. The following is a figurative but not exactly accurate summarization:
>
>         *   In the E-step, reference decisions for methodology selection are generated by exploring the expected utility of all methodologies. The methodology with the highest value is selected as the reference (Equation 7).
>
>         *   In the M-step, these reference decisions directly supervise the learning of the methodology-selection strategy.
>
>     *   **Efficient policy training.** EM facilitates efficient training of the methodology-selection sub-policy:
>
>         *   Standard RL explores only local regions within the policy space, relying on small policy-gradient updates.
>
>         *   EM uses a reference methodology-selection strategy to guide policy improvements, substantially pruning the policy search space.
>
>     *   **Theoretical Guarantees:** EM enjoys convergence guarantees under its theoretical foundation.

---

> ### Author Response · Authors · 2024-11-17
> **Rebuttal by Authors (Part2)**
>
> **2. Response to the weakness comments on Efficiency Assumptions, Q2 and Q3.**
>
>
> The reviewer shows doubts on the efficiency assumptions that EM narrows the policy search space, and raises question on how EM improve efficiency in methodology-selection.
>
> We realize that **there are misconceptions** that may be caused by the misunderstanding of "narrowing policy search space". So below we clarify this misunderstanding and that we do not make assumptions and do not claim EM improves the speed of methodology-selection.
>
> ----------------
> _(a) Clarifications on "narrowing policy search space"._
>
> In line 431, we stated that “the reference methodology-selection strategy helps narrow down the policy search space during training.” By this, _**we meant that EM prunes the hypothesis set of policy parameters for a better policy within the policy parameter space**_, according to \[1-3\]. This does not imply any assumption about computational load or speed improvements but rather refers to _**improved training efficiency**_.
>
> => _Q2: How does EM uniquely contribute to the narrowing of the policy search space?_
>
> The proposed EM uniquely "narrows policy search space", because _**it uniquely factorizes out the critical sub-policy, methodology-selection, and leverage reference decisions to focus policy improvement**_ (i.e., policy search) _**in regions with higher expected rewards.**_
>
> *   During the E-step, we compute reference decisions that are more likely to lead to a correct response.
>
> *   During the M-step, these decisions assist policy search in high-reward regions, and less promising areas of the policy search space are pruned.
>
>
> This mechanism parallels supervised learning approaches, where supervision accelerates convergence by providing direct guidance. Unlike standard Supervised Fine-Tuning (SFT), which relies on externally collected expert demonstrations, our EM approach generates reference methodology-selection decisions internally by harnessing the capabilities of the LLM itself. This internal guidance is a key differentiator that allows EM to uniquely narrow the policy search space.
>
> ----------------
> *(b) Addressing Assumptions and Claims about Efficiency and Computational Load in Methodology Selection*
>
> => _Q2: Could you provide empirical evidence or a theoretical basis for how EM improves efficiency in selecting between Chain of Thought (CoT) and code-based approaches?_
>
> => _Q3: What is the theoretical or experimental basis for assuming that EM will reduce computational load or enhance efficiency in methodology selection?_
>
> We regret the misunderstanding of making any assumptions and would like to emphasize that _**the argument of "narrowing policy search space" is mentioned as a conjecture**_ (c.f., line 430) _**based on empirical results, rather than an assumption without proof**_:
>
> *   In Section 3.2 (line 430), we explicitly frame the "narrowing policy search space" argument as a conjecture informed by empirical results in Figure 4 and Table 1.
>
> *   Figure 4 demonstrates that under identical training conditions, EM achieves superior performance at the same iteration compared to standard RL.
>
> *   Based on logical rationales (line 430) and empirical analysis (line 419), we reach at the conjecture for these results.
>
>
>
> **References**
>
> \[1\] Deisenroth M P, Neumann G, Peters J. A survey on policy search for robotics\[J\]. Foundations and Trends® in Robotics, 2013, 2(1–2): 1-142.
>
> \[2\] Levine S, Koltun V. Guided policy search\[C\]//International conference on machine learning. PMLR, 2013: 1-9.
>
> \[3\] https://icml.cc/2015/tutorials/PolicySearch.pdf

---

> ### Author Response · Authors · 2024-11-17
> **Rebuttal by Authors (Part3)**
>
> **3. Response to weakness comments on basline limitations, effectiveness of autonomous decisions, Q4 of comparison with baselines in methodology-selection and Q5 of correlations between methodology-selection and final performance.**
>
> We sincerely appreciate the constructive suggestions to analyze methodology-selection and its connection to final performance. However, it is important to clarify that **_analysis of methodology-selection as a standalone policy is trikcy_**. This is because we do not train a separate policy that maps a math query directly to a methodology, nor do we have a definitive "golden" answer for methodology-selection. Instead, methodology-selection and solution-generation are conceptually factorized in this work to ease the difficulty of autonomous code integration. Despite this factorization, these two components share parameters within the same LLM and are not inherently separable.
>
> ---------
>
> => *Q4: comparisons with baselines in methodology-selection.*
>
> => *Q5: correlations between methodology-selection and final performance.*
>
> We fully agree that the suggested analysis helps convince the significance of AutoCode Training in the final accuracy. Despite the challenges in analysis mentioned above, we strive to give a reasonable analysis on the effectiveness of methodology-selection and its connection to the final performance. **_We will include this analysis in our revised manuscript._**
>
> We first detail the baselines and evaluation metrics used to answer the above questions.
>
> **Baselines.** We compare our approach with several models that natively support both code and Chain-of-Thought (CoT) responses for math queries: GPT-4, Mammoth-70B trained using Hybrid Instruction Tuning, and DeepseekMath-Instruct-7B trained using tool-integrated reasoning annotations.
>
> **Evaluation Metrics.**
>
> -   **Final Pass@1 Accuracy of the Complete Response**
>
>     -   "Autocode": The LLM autonomously decides the methodology. For baselines without AutoCode Training, a four-shot prompt template (Appendix D.3) is used.
>     -   "Code": The LLM is explicitly prompted to generate a code response. For GPT-4, a four-shot template is applied. For other baselines, we use their native zero-shot templates.
>     -   "CoT": The LLM is explicitly prompted to use CoT reasoning with native templates.
> -   **Autocode CodeRate and Improvement Over Best Dictation**
>
>     -   **CodeRate** reflects the reduction in code usage compared to dictated code prompting.
>     -   **Accuracy Improvement over best performance** of either CoT or code dictation reflects the improvement of AutoCode over the native dictated inference.
> -   **Methodology-Selection Accuracy and Its Connection to Final Accuracy**
>
>     -   Ground-Truth Labels: Methodology-selection is treated as a binary classification task. The classification label is derived by performing 10 Monte Carlo rollouts per query with controlled methodologies. The optimal methodology is chosen based on higher expected correctness.
>     -   Imbalanced Classification: Since model capabilities differ, reference decisions are imbalanced (e.g., GPT-4 strongly prefers CoT, with only 7.45% of queries requiring coding). We report **mean accuracy across CoT-preferred and code-preferred queries** (Selection mAcc), along with per-class accuracy (in the last 3 and 4 columns).
>     -   Connection to Final Accuracy: Correct methodology-selection does not always guarantee a correct response due to prompt context mismatches and greedy decoding. We report **Pass@1 accuracy within correct selections** to evaluate how proper methodology-selection directly contributes to correct responses.

---

> ### Author Response · Authors · 2024-11-17
> **Rebuttal by Authors (Part4)**
>
> |                           | Autocode Pass@1 Acc (%) | Autocode CodeRate (%) | Autocode Improvement over Best Dictation (%) | Selection mAcc (%) | Pass@1 Acc /w Correct Selection (%) | CoT Pass@1 Acc (%) | Code Pass@1 Acc (%) | CoT Selection Acc (%) | Code Selection Acc (%) |
> |---------------------------|-------------------------|-----------------------|----------------------------------------------|--------------------|-------------------------------------|--------------------|---------------------|-----------------------|------------------------|
> | GPT-4                     | 74.16                   | 10.8                  | -2.5                                         | 50.51              | 89.27                               | 76.66              | 72.22               | 100                   | 1.03                   |
> | Mammoth-70B               | 31.46                   | 83.64                 | -5.76                                        | 64.87              | 77.62                               | 16.66              | 37.22               | 39.08                 | 90.66                  |
> | DeepseekMath-Instruct-7B  | 45.72                   | 12.94                 | -11.54                                       | 64.70              | 77.77                               | 45.32              | 57.26               | 97.62                 | 31.79                  |
> | AutoCode4Math-Qwen2-7B    | 64.64                   | 48.9                  | **+4.68**                                    | **88.25**          | **96.35**                           | 51.92              | 59.94               | 89.16                 | 87.34                  |
> | AutoCode4Math-Deepseek-7B | 65.28                   | 33.22                 | **+5.04**                                    | **86.53**          | **94.21**                           | 43.82              | 60.24               | 78.17                 | 94.90                  |
>
> **Analysis.**
>
> -   **Effectiveness in Methodology-Selection.** AutoCode Training significantly improves methodology-selection accuracy, outperforming baseline models by over 20% (compare row 3 with row 5). In particular, GPT-4 exhibits low mean accuracy (50.51%) due to its strong bias for CoT. If GPT-4 intelligently selected methodologies, it could achieve an additional 7.62% gain in accuracy for queries requiring code responses.
> -   **Accuracy Improvements and Code Reduction.** AutoCode Training enables our Deepseek-based model to achieve a 5.04% accuracy improvement compared to the best dictated inference (code prompting) while reducing code usage by up to 66%. Similar trends are observed with the Qwen2Math base model. In contrast, baseline models experience substantial accuracy drops when attempting autonomous methodology-selection. For example, DeepseekMath loses 11.54% accuracy when self-selecting methodologies (row 3).
> -   **Connection Between Methodology-Selection and Final Accuracy.** Baseline models surpass random selection (50%) in methodology-selection accuracy but fail to improve final accuracy over best dictation. This is accredited to* **the gap between methodology-selection and final correctness***: better methodology-selection is not always accompanied with correct final response, due to the greedy decoding and mismatch in prompt context. However, ***our proposed EM-based joint training significantly bridge this gap***: it improves Pass@1 accuracy within correct selections, achieving rates as high as 95% (rows 4 and 5). This success is due to training the LLM with complete responses, optimizing both methodology-selection and final correctness jointly.

---

> ### Author Response · Authors · 2024-11-17
> **Rebuttal by Authors (Part5)**
>
> **4. Response to Q6: Could you expand on the offline RL baseline and provide details on the data, algorithm, and setup of this experiment?**
>
> We would like to **clarify that we include details on the data, algorithm and experimental setups in the paper and contain an anonymous code repo for reproducibility.**
>
> The current version of the paper summarizes the key features of the datasets and experimental setups in lines 341--373. Due to page limits, we have included further details in the appendix, specifically:
>
> -   Section B: Provides an algorithm diagram.
> -   Section C: Includes additional details about data statistics and experimental setups.
>
> We also release our data and code in an anonymous repository (linked in line 373: <https://anonymous.4open.science/r/AnnonySubmission-35F0>), ensuring full transparency and reproducibility.
>
> Regarding the reviewer's suggestion to compare our approach with other RL methods or preference-based approaches, we offer the following clarifications:
>
> -   **Scope of the Proposed Approach**:
>
>     -   The proposed EM approach is specifically designed to address challenges in autonomous code integration for mathematical LLMs.
>     -   It is not intended as a general solution for LLM reasoning, and ***we do not claim its broad applicability in this area.***
> -   **Comparative Analysis with General Methods**: Common alternatives for general LLM reasoning have been extensively benchmarked in prior research [1-3], which deviates from the primary goal of this paper. In addition, these research have shown that ***RL with rewards outperforms preference-based methods when verification-based outcome feedback is available***.
>
> We understand that conducting a comprehensive comparison with all potential solutions for AutoCode, as suggested, could strengthen the argument for our approach. However, this requires tremendous efforts, and we will work on this once we have resolved the key weaknesses in the current submission.
>
> **References**
>
> [1] Shao Z, Wang P, Zhu Q, et al. Deepseekmath: Pushing the limits of mathematical reasoning in open language models[J]. arXiv preprint arXiv:2402.03300, 2024.
>
> [2] Gulcehre C, Paine T L, Srinivasan S, et al. Reinforced self-training (rest) for language modeling[J]. arXiv preprint arXiv:2308.08998, 2023.
>
> [3] Ahmadian A, Cremer C, Gallé M, et al. Back to basics: Revisiting reinforce style optimization for learning from human feedback in llms[J]. arXiv preprint arXiv:2402.14740, 2024.

---

> ### Author Response · Authors · 2024-11-24
> **Looking forward to Further Feedback**
>
> Dear Reviewer nGHB,
>
>
> Thank you for your valuable comments and suggestions.**Your feedback has greatly helped us improve our work and prompted us to reflect deeply on the core aspects of our proposed method.**
>
> **We have carefully addressed your questions and made revisions according to your feedback. The revised manuscript has been uploaded, and we have summarized the refinements in the top-most [post](https://openreview.net/forum?id=QhjosARfay&noteId=nQdxk1NfaQ).**
>
> We understand that this is a particularly busy time, so we sincerely appreciate any time you can spare to review our revisions and provide further feedback on whether our responses address your concerns. If there are any additional comments, we will do our utmost to address them promptly.
>
> Best Regards,
>
> The Authors

---

> ### Author Response · Authors · 2024-11-25
> **Discussion Phase is Ending. We are Actively Looking forward to Your Feedback.**
>
> Dear Reviewer 3NjD,
>
> Thank you for your valuable feedback, which greatly helped enhance our work. We have carefully addressed your questions and revised the manuscript in line with your insightful suggestions. **As the deadline for the public discussion phase approaches, we are eager to hear your further feedback.**
>
> We understand this is a particularly busy time, and we sincerely appreciate any time you can dedicate to reviewing our revisions. Your feedback on whether our responses sufficiently address your concerns would be immensely helpful. Should you have additional comments, we will do our utmost to address them promptly.
>
> Thank you once again for your time and effort.
>
> Best regards,
>
> The Authors

---

> ### Author Response · Authors · 2024-11-28
> **Waiting for Your Further Feedback**
>
> Dear Reviewer #nGHB,
>
> ***We are deeply grateful for your detailed response and have carefully addressed each of your questions***:
> - We compare our approach with relevant alternatives and highlight its unique strengths.
> - We clarify the misconception regarding the "narrowing policy search space."
> - We conduct additional experiments to analyze the methodology selection and its connection to the final accuracy.
>
> Importantly, ***we have made significant improvements to the paper’s presentation.*** The revised manuscript has been uploaded for your review. We are pleased to note that Reviewer #uEHE and Reviewer #Pg3w have acknowledged the improved presentation and indicated that their concerns have been addressed.
>
> We look forward to your further feedback and would greatly appreciate any additional insights you might provide. We understand this is a busy time and are truly grateful for the time and effort you dedicate to reviewing our revisions. Your expertise is invaluable to us.
>
> Thank you once again for your time and effort.
>
> Best Regards,
>
> The Authors.

---

### Official Review · Reviewer_3NjD · 2024-11-03

**Soundness:** 3
**Presentation:** 2
**Contribution:** 3
**Rating:** 5
**Confidence:** 3

**Summary:**

The authors identify a major challenge in current training regimes where models are typically told explicitly whether to execute code. Addressing this, the paper introduces an innovative EM algorithm that allows models to choose their execution paths autonomously, either using CoT reasoning or code execution. The results are quite impressive, showing a 65.28% accuracy in mathematics-related tasks, alongside a 65% reduction in code execution. This advancement not only improves efficiency but also enhances the practical applicability of the model.

**Strengths:**

1. The performance of the proposed method is commendable. It even surpasses the SFT approach on the same set of math related queries, which indicates its effectiveness and potential for broader application.
2. The area of LLM reasoning with tool usage is an important topic.
3. This paper involves many in-depth analysis on the proposed method and different experiment settings.

**Weaknesses:**

The organization of the second section of the paper appears somewhat confusing. It could benefit from a restructuring to enhance readability and coherence. Specifically, introducing an overarching pipeline at the beginning of the section could provide a clearer framework. And the relationship between sections 2.2 and 2.3 with section 2.1 needs to be more explicitly defined to better understand the flow and integration of ideas.

**Questions:**

1. Given that the proposed method primarily requires math-related queries, has there been any exploration into having the model generate its queries autonomously? This could potentially lead to self-improvement paradigm, further enhancing its effectiveness and applicability.
2. Will the authors consider open-sourcing the code?
3. Have the authors tried larger size models? (e.g. 13B)

---

> ### Author Response · Authors · 2024-11-17
> **Rebuttal by Authors (Part1)**
>
> Dear Reviewer #3NjD,
>
> We sincerely thank you for your valuable feedback, which has significantly helped us improve our paper and highlighted promising research directions. Below, we address the specific points raised in your review.
>
> **1. Response to Weakness Comments on Presentation:**
>
> We sincerely appreciate the reviewer's valuable suggestions. We acknowledge that the current organization of Section 2, which follows the technical derivation of the proposed EM approach, may be difficult to comprehend. To enhance the manuscript's readability and coherence, we plan the following improvements:
>
> -   **Section 2.1**: We will focus on presenting the problem statement and emphasize the key challenge faced by AutoCode---namely, the lack of reliable supervision for methodology selection.
> -   **Section 2.2**:
>
>     -   We will begin with an overview of the high-level ideas addressing this challenge and provide an overarching pipeline closely aligned with Figure 2. This will offer readers a clearer understanding of the proposed method.
>     -   Next, we will describe the key components of the proposed method, emphasizing their motivations and high-level concepts. Detailed technical derivations will be moved to the appendix for readers seeking further depth.
>
> ***We have uploaded our revised manuscript and detail our revisions in the top-most post.***
>
> **2. Response to Q1 on exploring autonomous** **query** **generation.**
>
> The reviewer has highlighted an exciting research direction: the development of a self-evolving training mechanism where the model generates queries it performs poorly on and reinforces its abilities through targeted training. This idea is indeed compelling, as it could address a significant limitation in RL-based training for mathematical reasoning.
>
> Existing RL training tends to converge on high-reward responses, as dictated by the reward-maximizing objective. When the query set remains fixed during training, the model's improvements plateau, as most training responses already achieve high rewards. Introducing new, challenging queries could mitigate this limitation and drive further advancements.
>
> While this idea holds immense potential, we have not yet delved deeply into its exploration. Autonomous query generation could tackle critical challenges in the field, making it a topic deserving of focused research efforts—efforts that cannot be adequately addressed within the constraints of the rebuttal phase. Specifically:
>
> -   **Lack of Query-Generation Capabilities**: AutoCode Training is designed to produce math experts, not math teachers capable of designing high-quality queries.
> -   **Challenges in Generating High-Quality Queries**: The open question of how to generate high-quality and diverse queries remains unresolved. Existing techniques, such as Evol-Instruct, have limited effectiveness because our dataset collected from various public math datasets already incorporates such augmentations, e.g., MetaMath, WizardMath. To this end, developing a specialized teacher LLM capable of devising math queries catering to the student's weaknesses is likely necessary. However, this requires significant additional research, including defining training objectives for the teacher model and coordinating its interactions with the student model. Such efforts fall outside the scope of the rebuttal phase.
>
> Despite these challenges, we agree that this is a promising research direction and is worthwhile to explore in future work. We thank the reviewer for bringing this important idea to our attention.
>
>  **3. Response to Q2 on code release**
>
> We confirm that ***we have released our data and code***, as noted in line 373, via an anonymous repository: https://anonymous.4open.science/r/AnnonySubmission-35F0

---

> ### Author Response · Authors · 2024-11-17
> **Rebuttal by Authors (Part2)**
>
> **4. Response to Q3 on larger model experiments.**
>
> We have conducted experiments using the production-scale 72B Qwen2 model. On the MATH benchmark, we observed a 3.45% improvement in accuracy (from 71.2%) and a 47% reduction in code executions compared to the code dictation baseline. However, we did not include these results in the manuscript for the following reasons:
>
> 1.  **Reproducibility**: Production-scale training was conducted using the Megatron framework for our own commercial use, which we cannot release. The code we have made available is based on a public codebase adapted with DeepSpeed. This code cannot support training over 13B.
> 2.  **Generalizability Across Models**: Training 70B models is computationally prohibitive, and we lack the resources to conduct experiments on multiple base models. To demonstrate generalizability, we focused on experiments with specialized 7B math LLMs, which is consistent with the approach taken in most prior work.
> 3.  **Performance of Smaller Models**: We acknowledge potential concerns regarding the scalability of our method. However, we emphasize that smaller specialized models such as Qwen2-Math and Deepseek-Math perform competitively with general-purpose large models on challenging benchmarks. For instance, DeepseekMath-Instruct-7B lags behind Claude-3 Opus by only 4% on MATH.
> 4.  **No 13B open-source LLMs.** For latest models like Qwen2-Math, Deepseek-Math, and the LLaMA-3 family, only the 7B and 70B versions are open-sourced. Recent open-source LLMs released in 2024 typically do not include a 13B variant, as they offer similar performance to the 7B models, and training an additional size is not cost-effective. The latest 13B open-source LLM is CodeLLaMA, released in August 2023 (https://huggingface.co/codellama/CodeLlama-13b-hf), more than a year ago.
>
> Additionally, ***we emphasize that AutoCode training benefits from combining the accuracy of both CoT inference and code-integrated inference.*** By intelligently selecting a proper methodology according the model's capability, AutoCode-trained models are more likely to combine the correctness of both CoT and code. The key improvement of AutoCode comes from closing the gap between the model's performance and the accuracy of oracle methodology-selection. ***This gain is size-agnostic as long as there is a significant gap to the oracle-selection upper bound.***

---

> ### Author Response · Authors · 2024-11-24
> **Looking Forward to Further Feedback**
>
> Dear Reviewer 3NjD,
>
> Thank you for your valuable comments and suggestions.**Your feedback has greatly helped us improve our work and has inspired new research opportunities.**
>
> **We have carefully addressed your questions and made revisions in line with your suggestions. The revised manuscript has been uploaded, and we have summarized the refinements in the top-most [post](https://openreview.net/forum?id=QhjosARfay&noteId=nQdxk1NfaQ).**
>
> We understand that this is a particularly busy time, so we sincerely appreciate any time you can spare to review our revisions and provide further feedback on whether our responses address your concerns. If there are any additional comments, we will do our utmost to address them promptly.
>
> Best Regards,
>
> The Authors

---

> ### Author Response · Authors · 2024-11-25
> **The Discussion Phase is Ending and We Sincerely Seek Your Further Feedback.**
>
> Dear Reviewer 3NjD,
>
> Thank you for your valuable feedback, which not only helped improve our work but also inspired new research opportunities. We have carefully addressed your questions and revised the manuscript based on your constructive feedback. As the deadline for the public discussion phase approaches, we sincerely seek your further feedback.
>
> We understand this is a particularly busy time, and we sincerely appreciate any time you can dedicate to reviewing our revisions. Your feedback on whether our responses sufficiently address your concerns would be immensely helpful. Should you have additional comments, we will do our utmost to address them promptly.
>
> Thank you once again for your time and effort.
>
> Best regards,
>
> The Authors

---

> ### Author Response · Authors · 2024-11-28
> **Sincerely Seeking Your Further Feedback**
>
> Dear Reviewer #3NjD,
>
> We sincerely appreciate your thoughtful and insightful feedback. We have carefully addressed each of your comments as follows:
>
> - **Presentation**: We have followed your suggestions to restructure Section 2, resulting in improved coherence and readability. We are pleased that ***Reviewer #uEHE and Reviewer #Pg3w have also acknowledged the enhanced presentation in the revised manuscript***.
>
> - **Autonomous Query Generation**: Your feedback on the open question regarding autonomous query generation has been immensely valuable. It brought this brilliant idea to our attention, and we have elaborated on its potential in addressing a key challenge in this field.
>
> - **Code Release and Larger Models**: We clarified that the code and data have been released through an anonymous repository. While we have trained on Qwen2-72B, we chose not to include the results in the current manuscript due to the prohibitive computational resources required to comprehensively evaluate multiple 70B models.
>
> ***We are eager to hear any further feedback you may have. This rebuttal process has been highly rewarding, particularly because our discussions have sparked new ideas for potential research opportunities.*** However, we understand you may have a demanding schedule and are truly grateful if you can spare some time to review our revisions and rebuttal. Your feedback is invaluable to us.
>
> Thank you once again for your time and effort.
>
> Best regards,
>
> The Authors

---

### Official Review · Reviewer_Pg3w · 2024-11-04

**Soundness:** 3
**Presentation:** 2
**Contribution:** 2
**Rating:** 6
**Confidence:** 3

**Summary:**

The paper introduces a method to train an LLM to select between use code-as-a-tool or chain-of-thought-reasoning actions, allowing the model to do some degree of inference-time search between the two actions. The training process is instantiated as an off-policy RL problem without labels for intermediate steps, by maximizing the joint probability of picking the correctness-maximizing action and correct response, decomposed into an expectation step to estimate state-action-values (via random rollouts) and a maximization step to train on the reward-weighted actions. 7B models trained using this method show improvement over just 2 iterations on the GSM8k and MATH datasets.

**Strengths:**

Significance: The topic addressed in this paper is exciting, with the release of o1 the community is interested in methods that can improve inference time decision making for math, especially on smaller models.
1. The results on GSM8k and MATH are interesting, over just 2 training iterations AutoCode improves performance in a stable way over baseline RL (Figure 4).
2. The method only relies on input and outputs, not using human generated labels is useful for data bottlenecked researchers. Framing the RL actions as CoT vs. code generation addresses the gap between self-learning the optimal policy instead of behaviorally cloning (SFT) a given policy as with prior models.

**Weaknesses:**

Performance:
1. Assuming I followed the table in Figure 3 correctly, it seems this method does not outperform the Qwen2-Math 7B-instruct model on any math dataset, which uses a simpler training method. The authors can try to see if their method improves Qwen2-Math 7B-instruct further. The method also doesn't give significant performance gains outside of MATH and GSM8k datasets over open-source baselines.

Presentation:
1. Section 2 is difficult to follow, it should be organized into clearer components by aligning better to Figure 2, highlighting (1) how the data is generated (expectation) and (2) how the model is trained on that data (maximization). I summarized the paper above based on my understanding, please clarify if there is misalignment.
2. The main results, Figure 3, looks like a spreadsheet screenshot, please re-organize it. I am not sure I followed the results of this figure properly (see weaknesses in performance section). Figure 4 needs to be larger.
3. Several typos, Figure 1: "sythesis" and Appendix B "referece"

**Questions:**

1. If you extend the AutoCode training to more iterations (Figure 4), does the performance on GSM8k and MATH continue to improve? It looks like the accuracy is still going up.
2. Why is the initial accuracy for Qwen-2-Math on GSM8k and MATH very different between Figure 3 and Figure 4?
3. I can't view the anonymized repo, can you double check it works? I would like to see the code and try the final model.

---

> ### Author Response · Authors · 2024-11-18
> **Rebuttal by Authors (Part1)**
>
> Dear Reviewer #Pg3w,
>
> We thank you for your valuable feedback in improving our work. We have carefully addressed each of your questions and detailed our responses below.
>
> **1. Response to Comments on Performance Weaknesses**
>
> The reviewer notes that AutoCode4Math-Qwen2 does not outperform Qwen2-Math-Instruct. ***However, we emphasize that this comparison is not conducted on a fair basis due to fundamental differences in their data quality:***
>
> -   **Qwen2-Math-Instruct**: This model benefits from private supervised fine-tuning (SFT) data, incorporating annotations from both human experts and GPT-4. Its improvement over Qwen2-Math-Base is driven by distillation from high-quality expert annotations.
> -   **AutoCode4Math-Qwen2**: In contrast, our model is trained on Qwen2-Math-Base using only publicly available mathematical queries paired with gold answers. The improvements observed in AutoCode4Math-Qwen2 stem entirely from the proposed algorithm, without leveraging any private data.
>
> To ensure a fair evaluation of the algorithmic contributions, we compare **AutoCode4Math-Qwen2** with **AutoCode4Math-RL** (see Table 1 and Figure 4). Both models use the same public query set and self-generated responses, thereby isolating the effect of expert annotations and focusing solely on the algorithmic enhancements.
>
> We understand the reviewer's concern on whether the proposed method can improve on top of SFT-ed model. However, we emphasize that ***we have validated the effectiveness of our proposed method on instruction-tuned models*** by experimenting with **Deepseek-Math-Instruct** (refer to lines 400--408), and witnessed significant improvement. This demonstrates the joint effect of the proposed method and high-quality private data.
>
> In the current paper, ***we do not train on top of Qwen2-Math-Instruct, because it does not natively support math coding abilities*** (see line 363).
>
> -   Unlike Deepseek-Math-Instruct, which natively support code integration for math reasoning, Qwen2-Math-Instruct does not incorporate Program-of-Thought or Tool-Integrated-Reasoning data in its training. To address this, ***additional fine-tuning*** would be necessary to enable math-coding capabilities. However, this process ***is constrained by the limitations of available public data***, which, as demonstrated, ***degrades performance***. For example, while fine-tuning with public data imparts some coding capabilities, it significantly reduces the performance of chain-of-thought (CoT) reasoning by more than 20% on MATH (see the comparison of bold and italic numbers in the table below).
> | Models                         | Prompt | GSM8K    | MATH     |
> |--------------------------------|--------|----------|----------|
> | Qwen2-Math-Instruct            | CoT    | **89.9** | **75.1** |
> | Qwen2-Math-Instruct-FurtherSFT | CoT    | _82.79_  | _52.98_  |
> | Qwen2-Math-Instruct-FurtherSFT | Code   | 85.02    | 58.98    |
>
> -   In addition, ***the instruction-tuning math query set used for Qwen2-Math-Instruct is proprietary***. Publicly available data falls significantly short in quality, limiting its effectiveness in fine-tuning.
>
>
> **2. Response to weakness comments on presentation**
>
> We regret the misunderstanding regarding "allowing the model to perform some degree of inference-time search between two actions" in the paper summary. To clarify, ***our proposed model decodes as standard LLMs do, without any search during inference.** *Instead, our approach can be seen as distilling search-based, high-reward trajectories into math LLMs during training. This enables the model to employ a more intelligent methodology-selection strategy during inference for better performance, but without the cost of increasing inference-time compute.
>
> We sincerely appreciate the reviewer's insightful suggestions and acknowledge that the current organization of Section 2, which follows a bottom-up technical derivation of the proposed EM approach, might hurt readability. To improve the manuscript's clarity and structure, we plan the following enhancements:
>
> -   **Section 2.1**: We will focus on clearly presenting the problem statement and emphasizing the key challenge faced by AutoCode -- specifically, the lack of reliable supervision for methodology selection.
> -   **Section 2.2**: We will adopt a top-down structure to present our approach:
>
>     -   We will start with a high-level overview of our proposed solution to this challenge and introduce a comprehensive pipeline aligned closely with Figure 2.
>     -   Following this, we will delve into the key components of our method, emphasizing their motivations and overarching concepts in the beginning. In particular, we will detail:
>
>         -   How the E-step generates the training data, and how we perform data synthesis.
>         -   How the M-step trains the model using this data in an efficient offline manner.
>
> *We will update our revised manuscript and detail our revisions in a separate comment.*

---

> ### Author Response · Authors · 2024-11-18
> **Rebuttal by Authors (Part2)**
>
> **3. Response to Q1 on extending AutoCode Training.**
>
> In discussing the performance growth of AutoCode Training, we would like to highlight an important aspect: ***the proposed approach leverages the "free lunch" of combining the accuracy of both CoT (Chain-of-Thought) reasoning and code-driven inference.*** Initially, certain queries are distinctively CoT-preferred or code-preferred due to the model's inherent capabilities: they cannot be solved by the alternative method. However, after AutoCode Training, the model learns to select the appropriate methodology for different queries. This enables the model to cover the correctness of both CoT-preferred queries and code-preferred queries. By developing a smarter methodology-selection strategy, AutoCode Training effectively bridges the gap to the performance of oracle selection.
>
> Thus, ***the convergence of AutoCode Training largely depends on the gap between the current model's performance and the upper bound of oracle selection.*** To illustrate this, we present three key metrics in the tables below:
>
> -   **Pass@1**: The accuracy of greedy decoding.
> -   **Oracle Pass@1**: The combined accuracy of CoT and code-driven inference. A query is considered correct if either CoT or code successfully solves it.
> -   **Selection mAcc**: The average accuracy of the learned methodology-selection strategy compared to the oracle selection strategy.
>
> Table 1. Performance Growth of AutoCode Training on MATH
> |                        | Pass@1 (%)                 | Oracle Pass@1 (%) | Selection mAcc (%) |
> |------------------------|----------------------------|-------------------|--------------------|
> | Deepseek-Math-Instruct | 45.32 (CoT);  57.26 (Code) | 67.14             | -                  |
> | AutoCode4Math-Iter1    | 62.84                      | 67.44             | 80.04              |
> | AutoCode4Math-Iter2    | 65.28                      | 67.98             | 86.53              |
> | AutoCode4Math-Iter3    | 66.38                      | 68.58             | 88.75              |
>
> Table 2. Performance Growth of AutoCode Training on GSM8K
> |                        | Pass@1 (%)                | Oracle Pass@1 (%) | Selection mAcc (%) |
> |------------------------|---------------------------|-------------------|--------------------|
> | Deepseek-Math-Instruct | 81.27 (CoT); 84.46 (Code) | 89.14             | -                  |
> | AutoCode4Math-Iter1    | 86.58                     | 90.66             | 75.43              |
> | AutoCode4Math-Iter2    | 89.39                     | 90.83             | 89.86              |
> | AutoCode4Math-Iter3    | 89.62                     | 90.97             | 91.41              |
>
> As shown in the above tables,
> - AutoCode Training yields minor improvements at iteration 3: +1.1% on MATH and +0.2% on GSM8K.
> - AutoCode Training yields continuous performance growth because the selection accuracy continues to grow, implying that the model learns continuously better methodology-selection.
> - This gain starts to converge because the upper bound of oracle selection converges. For example, Oracle Pass@1 67.44->67.98->68.58 on MATH, and the model performance has approached 66.38% leaving very small room for potential improvement. If we have access to queries with better quality, the potential room for improvement can be further opened up.
>
> **4. Response to Q2 on Figure 4 Initial Accuracy.**
>
> ***We have verified that the initial accuracy in Figure 4 is correct***, reflecting the CoT performance of the supervised fine-tuned (SFT-ed) model. The confusion may have arisen because Figure 3 reports only the code performance for *Code4Math-Qwen2*, rather than the CoT result, leading to a perceived misalignment.
>
> We clarify the numbers and explain how Figure 4 and Figure 3 align as follows:
>
> -   **Deepseek-Math-Instruct** achieves 81.27% on GSM8K and 45.32% on MATH with CoT, consistent with the result reported in Figure 3 (line 400).
> -   **Code4Math-Qwen2**, fine-tuned from Qwen2Math-Base using public math coding data, achieves 81.58% on GSM8K and 52.68% on MATH with CoT, and does not degrade CoT performance compared to its base model's performance of 80.74% on GSM8K and 51.82% on MATH. The reason why we do not train on top of Qwen2-Math-Instruct has been discussed in Response 1.
>
> We apologize for any confusion regarding Figure 4. For clarity, we will include CoT results in the revised manuscript.
>
> **5. Response to Q3 on code release**
>
> ***We confirm that our released code is accessible*** through the anonymous repository, as mentioned in line 373: https://anonymous.4open.science/r/AnnonySubmission-35F0

---

> > ### Comment · Reviewer_Pg3w · 2024-11-22
> > **Reviewer response**
> >
> > Thank you to the authors for responding in such detail!
> >
> > 1. It makes sense that you cannot apply the method on top of Qwen2-Math-Instruct as it does not have coding abilities. Thanks for clarifying the baseline may not be the best comparison. I would not debate the performance of your method further, they are interesting.
> >
> > 2. I would like to clarify the **greatest weakness of this paper is presentation**. Even if the research is great, it is hard to follow it and judge the contribution. Although the authors have clarified my questions in rebuttal, these details should be clear to follow in the paper itself.
> >
> > - Particular note: Please fix your figures, especially figure 3. A screenshotted spreadsheet is not an acceptable form of presentation.
> >
> > I am open to raising my score once the presentation of this work is improved.

---

> > > ### Author Response · Authors · 2024-11-24
> > > **Thank You for Your Support and We Uploaded the Revision**
> > >
> > > Dear Reviewer #Pg3w,
> > >
> > > We greatly appreciate your suggestions in improving the current manuscript and learned a lot from these valuable feedback.
> > >
> > > We have uploaded the revised manuscript. The revisions are marked in red in the new pdf, and we summarize the revisions in the top-most post (https://openreview.net/forum?id=QhjosARfay&noteId=nQdxk1NfaQ).
> > >
> > > Please let us know if you have any concerns left after our revision. We would be happy to discuss any further questions and comments you may have.
> > >
> > > Thank you once again for your feedback.
> > >
> > > Best regards,
> > >
> > > The authors of Submission 5899

---

> > > > ### Comment · Reviewer_Pg3w · 2024-11-25
> > > > **Thanks for fixing the presentation!**
> > > >
> > > > Thanks a lot to the authors for fixing the presentation of the paper and detailing every change. It is much better. I have raised my score accordingly.

---

> > > > > ### Author Response · Authors · 2024-11-26
> > > > > **Thank You for Raising Your Score**
> > > > >
> > > > > Dear Reviewer,
> > > > >
> > > > > Thank you for raising your score. We are greatly encouraged by your feedback and deeply appreciate your recognition of our work. We also sincerely thank you for taking the time to review our revisions, especially during this particularly busy period.
> > > > >
> > > > > Best Regards,
> > > > >
> > > > > The Authors

---

### Official Review · Reviewer_uEHE · 2024-11-04

**Soundness:** 3
**Presentation:** 2
**Contribution:** 2
**Rating:** 8
**Confidence:** 4

**Summary:**

In this paper, the authors address the problem of letting LLMs autonomously decide which planning strategy to use, between plain CoT and interaction with a python interpreter, in order to reliably solve math and reasoning problems. The authors correctly observe that such autonomic decision making cannot be induced via simple SFT on expert-annotated data, and therefore propose an RL strategy based on the EM algorithm, as a way to allow the model to self-explore and train itself on its own generated trajectories.

**Strengths:**

- The authors observation that autonomous code integration cannot be taught via simple SFT is definitely correct.
- The authors consider also multi-turn interactions featuring an interleaving of CoT and Python code execution, which is not found very often in the CoT and code execution literature.

**Weaknesses:**

The main weakness I can see is that it is not quite clear why the authors' EM approach is necessary. While I do agree that SFT is not sufficient for the task at hand, there are plenty of much simpler alternatives to the authors' approach, which would equally allow self-exploration and on-policy self improvement. Given how complex the authors' framework is, I feel that it is even more necessary to outline its advantage and convince readers that it is indeed worthwhile and preferable to simpler methods.

More points I wish to make:

- The author's outline of the method is extremely long and technical, and difficult to digest for any reader not already familiar with EM. Focusing more on its high-level idea and providing at least a few examples would go a long way.
- The main result table in figure 3 is honestly difficult to parse. Some lines are color-coded in blue and red, but the text and caption do not explain what this means. It's not clearly stated which lines correspond to the authors' proposed approach and which correspond to baselines. I personally would focus on comparing the author's proposal with whatever method happens to be the SOTA for the benchmarks the authors consider, in a much smaller table.
- Some typos can be found here and here, including in the abstract.

**Questions:**

- The authors' EM approach appears very complicated and requires a lot of moving parts, and is intended as a replacement for simple SFT. There are much simpler and proven ways of replacing SFT, such as Expert Iteration and RLHF, which can provide similar advantages in terms of self-exploration. Could the authors more clearly outline why their EM approach is preferable to these alternatives, and which advantage exactly is provided by each of its components?
- In section 2.3, the authors outline a framework for data synthesis. Fine-tuning LLMs on synthetically generated data is a proven technique for enhancing their abilities. What benefit do the other component of the EM framework bring?
- What exactly is the "standard RL" baseline mentioned in section 3.2? The authors should be a bit more specific.
- Again in section 3.2, the authors state "We conjecture that the proposed EM framework outperforms standard RL because: the
reference methodology-selection strategy helps narrow down the policy search space during training". Can the authors provide any experimental evidence for this statement?

---

> ### Author Response · Authors · 2024-11-15
> **Rebuttal by Authors (Part1)**
>
> Dear reviewer #uEHE,
>
> We sincerely appreciate your valuable feedback and suggestions, which have significantly helped us improve the clarity and presentation of our work. We have carefully addressed each of your questions and detailed our responses below.
>
> **1. Response to weaknesses: Why is the EM Approach Necessary?**
>
> The motivation for our EM framework lies in the need for math LLMs to autonomously develop effective methodology-selection strategies that complement their inherent strengths (line 059). The central challenge is the lack of reliable supervision for methodology-selection (line 182). Supervision based on externally dictated decisions fall short because it fails to dynamically adapt to the model's unique capabilities.
>
> We address this challenge by reformulating the problem as a maximum likelihood estimation with latent variables, naturally giving rise to an theoretically-sounded EM solution. To make it easier to understand why the proposed ***EM approach directly addresses the challenge of lacking reliable supervision***, here we give a figurative but not exactly accurate summarization of the proposed EM approach. The EM alternates between two steps:
>
> -   **E-step**: Generate reference decisions for methodology selection by exploring all methodologies and choosing the one with the highest value (Equation 7).
> -   **M-step**: Use these reference decisions for supervised fine-tuning (SFT) of methodology selection while reinforcing the solution-generation policy through offline exploration.
>
> Unlike SFT used in previous methods that resort to externally dictated methodology decisions, the reference decisions are obtained internally by monte carlo rollouts of the current solution-generation policy, which reflects the current model's unique strengths.
>
> Additionally, in response to the reviewer’s comment on the complexity of the proposed method, we emphasize that ***our approach presents a streamlined and efficient implementation of the EM framework*** (as summarized in the algorithm diagram in Section B). This makes our method highly practical and advantageous in real applications. We regret any misunderstanding that may have arisen and will elaborate on how to improve the presentation of our method in a separate rebuttal comment.

---

> ### Author Response · Authors · 2024-11-15
> **Rebuttal by Authors (Part2)**
>
> **2. Response to Q1: Comparison with Relevant Alternatives and Its Advantages**
>
> We first summarize the proposed method and alternatives that facilitate self-exploration and can replace the simple SFT.
>
>  - **The proposed EM approach** alternates between two steps. In the E-step, we generate reference decisions for methodology-selection. In the M-step, we leverage the reference decision to supervised the methodology-selection strategy, and reinforce the solution-generation policy with exploration.
>
>  - **RLHF/RL.** This type of algorithms reinforce the LLM language generation policy with exploration, based on the correctness feedback in the context of math reasoning. The exploration is typically done on-policy in standard RLHF methods such as instructGPT[1], which uses PPO, and ReST[2], which uses off-policy RL.
>
>  - **Expert Iteration[3] (ExIT)** alternates between two steps: expert improvement and policy distillation. During expert improvement, expert demonstrations are sampled by improving the current policy through some search procedures, such as monte-carlo tree search used in ExIT. During policy distillation, the expert demonstrations are used to supervised learning the current policy.
>
> The following table highlights the key differences between the EM framework, RLHF, and ExIT:
>
> | Aspect              | RLHF                                                    | ExIT                                                                    | Proposed                                                                                                         |
> |---------------------|---------------------------------------------------------|-------------------------------------------------------------------------|------------------------------------------------------------------------------------------------------------------|
> | Training Data       | Exploration                                             | Monte Carlo Tree Search to generate expert demonstrations               | Simple Monte Carlo Rollouts for reference methodology-selection decisions & exploration on solution-generation   |
> | Reinforcement       | Reinforce the full policy                               | None (improve via SFT)                                                  | Reinforce solution-generation policy                                                                             |
> | Supervised Training | None                                                    | Supervised training on expert demonstrations                            | Supervised Training on reference methodology-selection decisions                                                 |
> | Efficiency          | Training with standard exploration is often inefficient | Data generation is search-Intensive. Supervised training is efficient.  | Data generation requires only monte carlo simulations. Offline training without online exploration is efficient. |
>
> **3. Response to Q2: Benefits of each component of the proposed approach**
>
> We summarize the benefits of each component of the proposed EM approach below.
> - **The E-step** benefits from a proven way to optimally generate reference decisions for methodology-selection (Equation 4), which provides effective guidance for improving the current methodology-selection strategy. This **_directly addresses the challenge of lacking reliable supervision for methodology-selection_**: (a) the reference decisions are used to supervise the learning of methodology-selection in the M-step; (b) the reference decisions are provably reliable, because they maximizes the expected utility of the current solution-generation policy, as shown in Equation 4. In practice, these reference decisions are generated by exploring the model's inherent capability. In simple (but not accurate) words, we try all decisions, observe the outcomes, and set the decision that yields highest value as the reference decision.
> - **The M-step** benefits from the supervised training of methodology-selection with the reference decisions and **_reinforcement learning with exploration_** to improve the solution-generation policy.
> - **Efficient Joint Training** benefits **_training methodology-selection and solution-generation in a joint and efficient manner_**. Indeed, methodology-selection and solution-generation share the parameters in a single LLM policy, so it is not reasonable to separate the training of methodology-selection and solution-generation. To align the training condition of solution-generation with the supervised training of methodology-selection, we thus adopt off-policy RL, which allows us to RL with an offline curated exploration data. The algorithm is efficient as the LLM policy is trained in a fully offline manner, compared to on-policy RL methods.

---

> ### Author Response · Authors · 2024-11-15
> **Rebuttal by Authors (Part3)**
>
> **4. Response to Q3: Clarification on "the standard RL" baseline**
>
> In our work, "standard RL" refers to reinforcement learning with correctness feedback in the context of math reasoning. "RLHF" typically refer to reinforcement learning with human feedback to reinforce the general LLM generation. Therefore, **_"the standard RL" baseline essentially refers to using the same RL technique in "RLHF" but with correctness feedback in the context of math reasoning._**
>
>
> **5. Response to Q4: Evidence for the conjecture why EM outperforms RL**
>
> We refer to narrowing policy search space as  reducing the set of promising hypotheses of better policy within the policy space. For instance, learning from expert demonstrations narrows the policy search space compared to random exploration because the expert demonstrations provide direct guidance on improving the current policy, thus constraining the set of promising hypotheses.
>
> As mentioned above, the M-step in the proposed EM approach involves supervised learning for methodology-selection, which resembles learning from expert demonstrations. However, a key distinction is that the demonstrations—the reference decisions—are derived from the model's internal exploration of its unique strengths rather than relying on externally dictated decisions.
> Logically, this supervised procedure takes similar effect to learning from demonstrations, so we conjecture that the proposed EM narrows the policy search space.
>
> **We present empirical evidence from two perspectives:***
>
> - Policy Improvement: EM achieves a better policy under identical training conditions and at the same iteration compared to RL. This is evident from the convergence curve shown in Figure 4.
> - Methodology-Selection Alignment: EM demonstrates better alignment between its improved methodology-selection strategy and the optimal strategy.
>
> The table below summarizes these findings. It compares EM and RL training with the base SFT-ed LLM policy they are initialized from. We also include the conceptual oracle strategy for reference, which selects the optimal methodology for each query assuming fixed solution-generation (regardless of the improvement on solution-generation). We evaluate two metrics on the MATH benchmark: (a) Pass@1 Accuracy: The accuracy of greedy decoding the whole response. (b) Alignment Accuracy: The alignment of the methodology-selection strategy with the oracle selection strategy.
>
> | Metrics                         | AutoCode4Math-EM@Iter1 | AutoCode4Math-RL@Iter1 | Base-SFT@Iter0 | Oracle |
> |---------------------------------|------------------------|------------------------|----------------|--------|
> | DeepseekMath Pass@1 Acc (%)     | 61.08                  | 56.1                   | 45.32          | 66.74  |
> |     Selection Alignment Acc (%) | 69.77                  | 62.73                  | 52.88          | 100    |
> | Qwen2Math Pass@1 Acc (%)        | 63.88                  | 61.26                  | 52.84          | 67.36  |
> |     Selection Alignment Acc (%) | 81.38                  | 76.69                  | 73.78          | 100    |
>
> As shown in the above table, the methodology-selection strategy trained using EM aligns significantly better than using RL, across different base models: +7% for DeepseekMath, and +4.6% for Qwen2Math. This drives the improvement on pass@1 accuracy of the whole response: +5% for DeepseekMath, and +2.6% for Qwen2Math. These results show that EM better aligns the methodology-selection with the oracle strategy, implying that EM finds a better policy in the policy search space using the reference decisions. Notably, EM achieves closer gap with the oracle pass@1 accuracy: 5.6% gap for DeepseekMath, and 3.5% gap for Qwen2Math.  Therefore, we conclude that EM outperforms RL because it narrows the policy search space.
>
> **References:**
>
> [1] Ouyang, Long, et al. "Training language models to follow instructions with human feedback." Advances in neural information processing systems 35 (2022): 27730-27744.
>
> [2] Gulcehre, Caglar, et al. "Reinforced self-training (rest) for language modeling." arXiv preprint arXiv:2308.08998 (2023).
>
> [3] Anthony, Thomas, Zheng Tian, and David Barber. "Thinking fast and slow with deep learning and tree search." Advances in neural information processing systems 30 (2017).

---

> ### Author Response · Authors · 2024-11-24
> **Looking forward to Further Feedback**
>
> Thank you for your insightful comments and suggestions. **Your feedback has been instrumental in improving our work and has prompted us to reflect deeply on the core aspects of our proposed method.**
>
> **We have carefully addressed your questions and incorporated revisions based on your feedback. The revised manuscript has been uploaded, and we have summarized the refinements in the top-most [post](https://openreview.net/forum?id=QhjosARfay&noteId=nQdxk1NfaQ).**
>
> We understand that this is a particularly busy time, so we sincerely appreciate any time you can spare to review our revisions and provide further feedback on whether our responses address your concerns. If there are any additional comments, we will do our utmost to address them promptly.
>
> Best Regards,
>
> The Authors

---

> ### Author Response · Authors · 2024-11-25
> **The Discussion Phase is Ending and We Eagerly Seek Your Feedback**
>
> Dear Reviewer uEHE,
>
> Thank you for your valuable feedback, which greatly helped us improve our work. We have carefully addressed your questions and revised the manuscript based on your constructive suggestions. **As the deadline for the public discussion phase approaches, we are eager to hear your further feedback.**
>
> We understand this is a particularly busy time, and we sincerely appreciate any time you can dedicate to reviewing our revisions. Your feedback on whether our responses sufficiently address your concerns would be immensely helpful. Should you have additional comments, we will do our utmost to address them promptly.
>
> Thank you once again for your time and effort.
>
> Best regards,
>
> The Authors

---

> > ### Comment · Reviewer_uEHE · 2024-11-26
> >
> > I thank the authors for their extensive and detailed rebuttal, as well as for their diligent revision of their paper taking into account mine and the other reviewers' concerns.
> > I have no further questions and the paper is much improved now. I will therefore raise my score to a straight Accept.

---

> > > ### Author Response · Authors · 2024-11-27
> > > **Thank You for Raising the Score**
> > >
> > > We sincerely thank the reviewer for raising the score!
> > >
> > > Most importantly, we deeply appreciate the time and effort dedicated to reviewing our paper and rebuttal. This process has provided us with a truly valuable opportunity to improve our paper and gain precious insights into how to effectively deliver our research to the audience.
> > >
> > > Thanks once again!

---

### Author Response · Authors · 2024-11-24
**Summary of Rebuttal by Authors**

Dear Reviewers and Area Chairs,

We sincerely thank all the reviewers for their constructive and thoughtful feedback. ***We are particularly encouraged by the following positive remarks:***

**Significance of the Autonomous Code Integration:**

- ﻿Reviewer **#uEHE**: "The authors observation that autonomous code integration cannot be taught via simple SFT is deìnitely correct"

- Reviewer **#Pg3w**: "﻿The topic addressed in this paper is exciting."

- Reviewer **#3NjD**: "﻿The area of LLM reasoning with tool usage is an important topic."

- Reviewer **#nGHB**: "﻿This research identiìes an important gap in current language models' ability to self-direct their reasoning strategies"

**Contribution and Novelty of the Proposed Method:**

- Reviewer **#nGHB**: "﻿The study makes a signiìcant contribution by addressing how models can autonomously choose between .."

- Reviewer **#Pg3w**: "﻿The method only relies on input and outputs, not using human generated labels is useful for data bottlenecked researchers."

- Reviewer ﻿**#uEHE**: "﻿The authors consider also .., which is not found very often in the CoT and code execution literature"

**Effectiveness and Soundness of the Proposed Method:**

- Reviewer **#nGHB**: "﻿Through their experiments, the authors demonstrate that .. leads to measurable performance improvements".

- Reviewer **#3NjD**: "﻿The performance of the proposed method is commendable."

- Reviewer **#Pg3w**: "﻿The results .. over just 2 training iterations AutoCode improves performance in a stable way"

﻿We deeply appreciate these positive comments and are grateful for the reviewers' recognition of our work.

Besides, we extend our sincere thanks for the valuable suggestions on improving the presentation of the manuscript. Based on this feedback, ***we have uploaded a revised version of the manuscript. We marked in blue the revisions in the uploaded pdf and summarized the revisions in the above post (https://openreview.net/forum?id=QhjosARfay&noteId=nQdxk1NfaQ)***.

In addition, ***we have carefully addressed each reviewer's questions individually***, including the following key points:

- Clarified the motivation and unique benefits of the proposed methods (per #uEHE, #nGHB's feedback).

- Conducted detailed comparisons with relevant alternatives (per #uEHE, #nGHB's feedback)

- Performed additional experiments to further analyze the performance convergence of the proposed approach (per #Pg3w's feedback)

- Provided an in-depth analysis to justify how the proposed method helps narrows the policy search (per #uEHE, #nGHB's feedback)

- Provided an in-depth analysis of methodology selection accuracy and its connection to final accuracy (per #nGHB's feedback)

***Reviewer #uEHE and Reviewer #Pg3w have acknowledged that their concerns have been addressed and recognized the improvements made in the paper.*** They have updated their scores to eight and six, respectively. We are currently awaiting further feedback from Reviewer #nGHB and Reviewer #3NjD.

We sincerely thank all the reviewers and the area chairs for their time, effort, and thoughtful insights throughout the review process. Your feedback has been invaluable in refining our work, and we deeply appreciate your contributions.

Best regards,

The authors of Submission 5899

---

### Author Response · Authors · 2024-11-24
**Summary of Paper Revision**

In response to the reviewers' suggestions for improving the presentation of the manuscript, we have made the following refinements:

**Restructuring Section 2 for Improved Readability and Coherence:**\
Following reviewer **#3NjD**'s feedback, we have reorganized Section 2 into three subsections:

1. Section 2.1 formally introduces the problem statement.

2. Section 2.2 presents the EM framework for solving this problem.

3. Section 2.3 provides a streamlined and efficient implementation of the EM framework.

**Refining Technical Derivations of the EM Framework:**\
Incorporating reviewer **#uEHE**'s advice to use examples and **#3NjD**'s suggestion of an overarching outline, we have made several enhancements:

-   At the start of Section 2.1, we clearly articulate the problem challenges (line 145) and use an analogy based on human cognitive processes to motivate the discussion (line 152, as advised by #uEHE).
-   We introduce an overarching pipeline (line 160, per #3NjD's suggestion) before diving into technical derivations.
-   The derivations are now more concise and focus on high-level ideas, addressing reviewer #uEHE's feedback.

**Adding Comparative Analysis with Relevant Alternatives:**\
Based on feedback from reviewers **#uEHE** and **#nGHB**, we have included a new paragraph discussing the advantages of our proposed approach compared to SFT and standard RL (line 226).

**Aligning Section 2 with the Figures:**\
In response to reviewer **#Pg3w**'s advice to better align section 2 with Figure 2, we do the following:
- Before detailing the EM framework, we summarize the high-level idea of the proposed method as self-exploration in E-step and self-refinement in M-step (line 180), which now aligns with the left side of the figure.
- In discussing the EM framework, we explain how the E-step identifies the reference strategy (line 212) and the M-step improves policy (line 223), which now aligns with the middle of the figure.
- In discussing the practical implementation of the EM framework (Section 2.3), we describe data curation and efficient joint training which now corresponds to the right side of the figure.

**Improving Figures 2, 3, and 4:**\
Following reviewer **#Pg3w**'s suggestions, we have refined the figures as follows:

-   Figure 2: Enhanced to better align with the revised manuscript's method overview.
-   Figure 3: Replaced with a LaTeX table for clarity, presenting baseline results and highlighting improvements achieved by our approach.
-   Figure 4: Adjusted for size and revised to remove occluded numbers for improved readability.

**Adding Experiments**

Based on the feedback of reviewer  **#uEHE** and **#nGHB**, we put the comparison on autonomous code integration with baselines in the Appendix Section C.

**Misc**

We highlight in the abstract that code and data is released via an annonymous repo, as we notice some reviewers do not find the repo we previously note in line 373.

Thank again for the reviewers' valuable suggestions, which greatly help us improve the manuscript.

---

### Meta-Review · Area_Chair_eYDF · 2024-12-24

**Metareview:**

In this paper, the authors address the problem of letting LLMs autonomously decide which planning strategy to use, between NL CoT and code generation / execution, in order to reliably solve math and reasoning problems. The author observe that such autonomic decision making cannot be induced via simple SFT on expert-annotated data, and therefore propose an RL strategy based on the EM algorithm, as a way to allow the model to self-explore and train itself on its own generated trajectories. They show improvements on GSM8k and MATH.

Reviewers generally liked the strengths of the reported results. All reviewers did not like how the method was presented in the paper and most mentioned not understanding the details in the EM and the ELBO bound. This paper treats using NL vs. code as the latent variable and then applied EM. While EM is a 60 year old method, it is not commonly used in LLM-training where neither steps are solvable. Reviewers nGHB and Pg3w questioned how this method should be compared and baselined against RL and if they were correctly compared with EM. After thinking about this, this is quite important since it is unclear why EM is more suitable at all since p(reward | latent c) can be estimated by rollouts. EM is needed when the latent variables cannot be observed, which does not seem to be the case here.

So this paper took a very simple concept (model picks which strategy to use), and results that seem to be strong (good improvements over the original baseline and over a RL baseline on 1+ round), but wrote it so that reviewers and I cannot figure out what's actually the main method and how can it be possibly be better than baseline (within a reasonable amount of time). Here are some reasons why I think it's not just due to us not understanding the authors properly:

* The authors included unnecessary details of how EM works like the ELBO bounds. The key to adding EM is just what is the latent variable and what are the parameters. They should highlight how EM can possibly be better.

* nGHB: The authors did not clearly identify what is their RL baseline or even what is the supervised baseline, since their EM also needs to do the rollouts and collect some sufficient statistics p(r | c, x). They could have clearly explained which part of the EM is not handled well by RL, but they did not. This is a very important point, since there are many possible RL baselines, some of which can easily model p(c | x) using Bayes rule, and p(r | c) with a reward model, bring it to feature parity with the EM method. There were no mentions or discussion of this nor is it included in the main result table.

* They do not control for or mention inference time compute so it is hard to interpret the results. For this paper, I think the empirical strength matter a lot. For instance, Deepseek actually gets 60%+ by aggregating over 64 samples. Using the tool integrated setting, Deepseek gets 86.7% on GSM8k and 58.8% on MATH. Their paper seems to fall under this tool-integrated setting. While under the RL COT, which is also similar to their setting, reporting 88.2% 51.7%.

Voting reject since the reviews were mixed and we should discourage papers that make a simple concept hard to understand and not presenting results against reasonable baselines. The authors are encouraged to simplify and highlight the key improvement if they believe in their method, instead of framing it as EM vs RL or SFT.

Reviewer uEHE voted accept after author rebuttal but does not explain why, so their review still reads like a reject. I took their opinion into account but cannot see how they advocate for acceptance.

**Additional Comments On Reviewer Discussion:**

authors managed to address many reviewers concerns. AC had a bit of discussion with the main advocating reviewer.

---

### Decision · Program_Chairs · 2025-01-22

Reject